# Noise-Augmented Deep Neural Networks for Image Classification: Insights from Information Theory

## Abstract

In this study, we explore the impact of proactively injecting noise into deep learning models, focusing particularly on image classification and domain adaptation. While noise is typically seen as harmful, our findings reveal that, under certain conditions, noise can beneficially influence the entropy of the system, enhancing the learning outcomes. We employ information entropy to characterize the complexity of the learning tasks and categorize noise into two types, positive noise (PN) and harmful noise (HN), based on whether it helps reduce task complexity. We theoretically prove that positive noise reduces task complexity and demonstrate the presence of positive noise through extensive experiments on Convolutional Neural Networks (CNNs) and Vision Transformers (ViTs). We further propose NoisyNN, an innovative approach to leverage positive noise. NoisyNN achieves state-of-the-art performance on various image classification and domain adaptation tasks. Extensive experiments conducted on 15 datasets, including popular image datasets and out-of-distribution datasets, demonstrate the efficacy of our method. Our study provides the community with a new paradigm for improving model performance. Our code is available at https://anonymous.4open.science/r/CodeBase-56B0.

## 1 Introduction

Noise, commonly viewed as an obstacle in machine learning and deep learning applications, is universal due to various factors such as environmental conditions, equipment calibration, and human activities Ormiston et al. (2020); Thulasidasan et al. (2019). In computer vision, noise can emerge at multiple stages. During image acquisition, for instance, camera sensors or other imaging devices may introduce noise. This could manifest as electronic or thermal noise, leading to random variations in pixel values or color discrepancies in the captured images Sijbers et al. (1996). Additionally, noise can also be introduced during the image preprocessing phase. Operations such as image resizing, filtering, or color space conversion are potential sources of noise Al-Shaykh & Mersereau (1998). Prevailing literature typically assumes that noise adversely affects the task at hand Sethna et al. (2001); Owotogbe et al. (2019). However, is this assumption always applicable? Our work seeks to thoroughly examine this critical question. We recognize that the vague definition of noise contributes to the uncertainty in identifying and characterizing it. One effective way to categorize different noises is through analysis of task complexity change (Li, 2022). Leveraging the concept of task complexity, we can categorize noise into two types: positive noise (PN) and harmful noise (HN). PN reduces task complexity, whereas HN increases it, consistent with traditional views of noise.

Our work, which combines a theoretical analysis based on information theory with extensive empirical evaluation, reveals that the *simple injection of noise into deep neural networks, when done in a principled manner, can significantly enhance model performance*. This study primarily examines three prevalent types of noise: Gaussian noise, linear transform noise, and salt-and-pepper noise. Gaussian noise is characterized by random data fluctuations following a Gaussian distribution. Linear transform noise involves affine elementary transformations applied to the data or embeddings. Salt-and-pepper noise introduces random black or white pixels to images or replaces some values of an embedding with its maximum or minimum values. **We show that both Gaussian noise and salt-and-pepper noise are harmful noise when injected into the latent features in the embedding space, while linear transform noise can be made positive noise under proper constructions.**

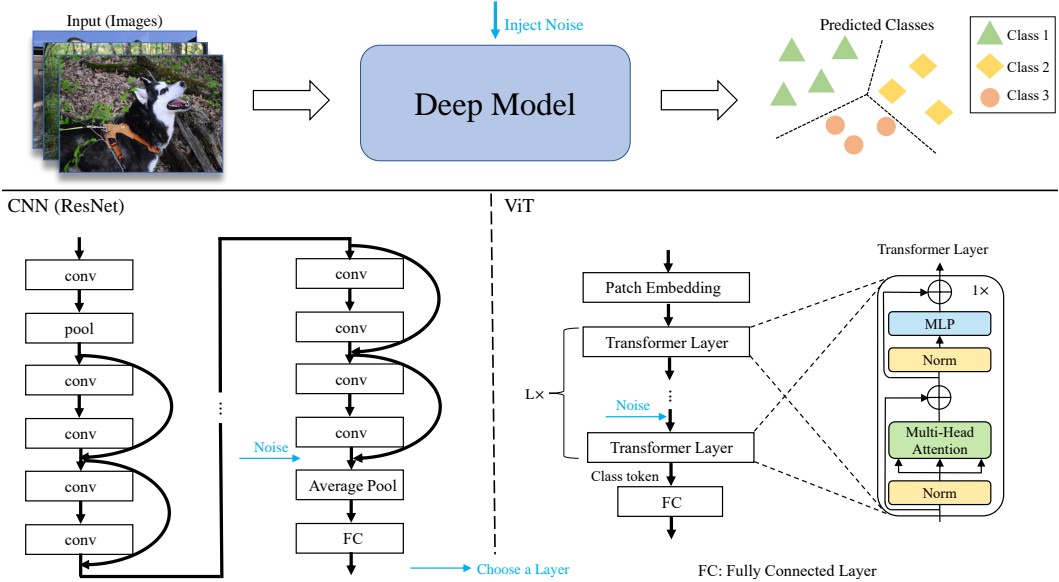

Figure 1: An overview of the NoisyNN framework. Showing the unified pipeline for image classification problems utilizing deep models such as CNNs or ViTs. The blue arrow indicates the injection of noise into the embeddings at the chosen layer.

Additional experiments with other noises such as dropout (Srivastava et al., 2014) further confirm the effectiveness of our proposed approach (App Table 19).

We start by presenting a comprehensive theoretical analysis of how these three types of noise impact deep learning models. Building on this theoretical foundation, we propose NoisyNN, a novel method designed to enhance the deep neural network performance on Image Classification and Domain Adaptation. We conduct extensive experiments with two prominent model families, Vision Transformers (ViTs) and Convolutional Neural Networks (CNNs), to validate the effectiveness of NoisyNN. Our empirical findings demonstrate the huge benefits of leveraging positive noise.

The contributions are summarized as follows:

- **First**, we re-examined the impact of different common noises on deep learning models. Our theoretical and empirical findings show that certain noise can enhance model performance.

- **Second**, we introduce NoisyNN, an innovative approach that utilizes positive noise. NoisyNN achieves state-of-the-art results on various image classification and domain adaptation tasks.

- **Third**, our study, along with the success of NoisyNN, prompts revisiting the role of noise in machine learning and opens new avenues for future research in leveraging noise.

## 2 RELATED WORK

**Positive Noise.** While noise is often assumed harmful to tasks, empirical evidence also suggests useful applications of noise (Li, 2022). In signal processing, it has been shown that random noise can facilitate stochastic resonance, enhancing the detection of weak signals Benzi et al. (1981). In neuroscience, noise has been recognized for its potential to boost brain functionality McClintock (2002); Mori & Kai (2002). In machine learning, the study of noise also draws a lot of interest (Kosko et al., 2020; Minsky, 1961; Bishop, 1995; Reed et al., 1995; An, 1996) with various applications spanning wide areas such as image classification (Li, 2022), Natural Language Processing (NLP) (Pereira et al., 2021; Khan et al., 2023), training generative adversarial networks (GANs) (Song & Ermon, 2019; Kim et al., 2024; Wang et al., 2023), and finetuning large language models (Jain et al., 2023b).

Recent work by Li (2022) marks a significant advance in the theoretical understanding of different noises. By employing information theory, they differentiate between beneficial "positive noise" and detrimental "pure noise", based on their impact on task complexity. However, their analysis has three notable limitations: 1. it is confined to *only the image space*; 2. all experiments are conducted *only on shallow models*, far from the current best practices 3. it does not answer the practical question: *how to create and leverage positive noise?* Our study aims to address these limitations. We answer the questions: *Does positive noise exist for deep models?* and if so, *how to leverage the positive noise?* We make a significant extension to the positive-noise framework in Li (2022). Our work not only confirms the presence of positive and harmful noise in embedding space but also finds that *leveraging positive noise in deeper layers of the embedding space is often more effective* (see Fig 2 b&d). Furthermore, we propose a practical approach to leverage the positive noise in deep models, we term it "NoisyNN". NoisyNN promises to unlock new potentials in the application of noise for enhancing neural networks. Other lines of work includes (Kosko et al., 2020; Adigun & Kosko, 2023), which take on an expectation maximization (EM) perspective.

**Data Augmentation** Data augmentation plays an important role in training deep vision models (Yang et al., 2023b). The general idea of data augmentation is to compose transformation operations that can be applied to the original data $x$ to create transformed data $x'$ without severely altering the semantics. Common data augmentation range from simple techniques like random flip and crop (Krizhevsky et al., 2012) to more complex techniques like MixUp (Zhang et al., 2017), CutOut (DeVries & Taylor, 2017), AutoAugment (Cubuk et al., 2019), AugMix (Hendrycks et al., 2019), RandAugment (Cubuk et al., 2020). More comprehensive reviews can be found in (Mumuni & Mumuni, 2022). Our approach is closely related to the research on data augmentation but stands apart due to its theoretical foundation. Our framework provides a more controlled and principled way to augment data, setting it apart from conventional methods, which often require substantial domain knowledge and ad-hoc design, as noted in (Cubuk et al., 2020). Later experiments show that our approach outperforms traditional data augmentation techniques (App Table 18) and is compatible with other data augmentation techniques (App Table 17).

**Comparison with Manifold MixUp.** Our NoisyNN shares some similarities with Manifold MixUp (Verma et al., 2019), a regularization technique designed for supervised image classification that extends the MixUp strategy to the embedding space by linearly interpolating embedding vectors $z_i$ (instead of images $x_i$) along with their corresponding labels $y_i$. However, there are several key differences. Unlike Manifold MixUp, which aims to flatten class representations through training on interpolated synthetic samples, our NoisyNN is grounded in a theoretical analysis of how noise injection impacts task entropy, as introduced by (Li, 2022).

Additionally, we derived the optimal form of noise injection (Eq.20) within the linear transform noise design space, which Manifold MixUp does not provide. Procedurally, Manifold MixUp interpolates both embeddings and labels to generate synthetic samples, followed by training on these samples, as its theoretical foundation relies on modifying both features and labels. In contrast, our method perturbs only the embeddings and leaves the labels unchanged, as our theoretical analysis is based on un-interpolated labels. Investigating whether label interpolation could be integrated into our theoretical framework may be a promising avenue for future research. Experiments in App Table 20 show the superior performance of NoisyNN. More comparison can be found in App F.8.

## 3 METHODS

In information theory, the entropy Shannon (2001) of a random variable $x$ is defined as:

$$H(x) = \begin{cases} -\int p(x) \log p(x) dx & \text{if } x \text{ is continuous} \\ -\sum_x p(x) \log p(x) & \text{if } x \text{ is discrete} \end{cases} \tag{1}$$

where $p(x)$ is the distribution of the given variable $x$. The mutual information of two random discrete variables $(x, y)$ is denoted as Cover (1999):

$$\begin{aligned} MI(x, y) =& D_{KL}(p(x, y) \parallel p(x) \otimes p(y)) \\ =& H(x) - H(x|y) \end{aligned} \tag{2}$$

where $D_{KL}$ is the Kullback–Leibler divergence Kullback & Leibler (1951), and $p(x, y)$ is the joint distribution. The conditional entropy is defined as:

$$H(x|y) = -\sum p(x, y) \log p(x|y) \tag{3}$$

These definitions can be extended to continuous variables by replacing the summation with integral.

Following Li (2022), we use $\mathcal{T}$ to denote the learning tasks of a deep model mapping from the dataset to the corresponding labels. Leveraging principles from information theory, we can quantify the complexity of the learning task $\mathcal{T}$ through information entropy $H(\mathcal{T})$ Li (2022). This approach allows us to gauge task difficulty, where lower entropy indicates an easier task, and vice versa. Denote the noise by $\epsilon$. The task complexity change when adding noise $\epsilon$ can then be measured (Li, 2022):

$$\triangle S(\mathcal{T}, \epsilon) = H(\mathcal{T}) - H(\mathcal{T}|\epsilon) \tag{4}$$

Formally, noise that reduces task complexity, i.e., $\triangle S(\mathcal{T}, \epsilon) > 0$, is defined as **positive noise** (PN). Conversely, **harmful noise** (HN) when $\triangle S(\mathcal{T}, \epsilon) \leq 0$.

$$\begin{cases} \triangle S(\mathcal{T}, \epsilon) > 0 & \epsilon \text{ is positive noise} \\ \triangle S(\mathcal{T}, \epsilon) \leq 0 & \epsilon \text{ is harmful noise} \end{cases} \tag{5}$$

### 3.1 INFLUENCE OF DIFFERENT NOISES ON TASK ENTROPY

We provide a general framework to analyze the influence of different noises on the classification tasks with the CNNs and ViTs backbones. The framework is depicted in Fig. 1. By injecting specific noise under certain conditions into the embeddings of an intermediate layer, a model has the potential to gain additional information to reduce task complexity, thereby improving its performance.

In classification problems, the dataset $(\boldsymbol{X}, \boldsymbol{Y})$ can be regarded as samples from $D_{\mathcal{X}, \mathcal{Y}}$, where $D_{\mathcal{X}, \mathcal{Y}}$ is some unknown joint distribution of data and labels from feasible space $\mathcal{X}$ and $\mathcal{Y}$, i.e., $(\boldsymbol{X}, \boldsymbol{Y}) \sim D_{\mathcal{X}, \mathcal{Y}}$ Shalev-Shwartz & Ben-David (2014). Hence, given a set of $k$ data points $\boldsymbol{X} = \{X_1, X_2, ..., X_k\}$, the label set $\boldsymbol{Y} = \{Y_1, Y_2, ..., Y_k\}$ is regarded as sampling from $\boldsymbol{Y} \sim D_{\mathcal{Y}|\mathcal{X}}$. The complexity of $\mathcal{T}$ on $\boldsymbol{X}$ is formulated as:

$$H(\mathcal{T}; \boldsymbol{X}) = H(\boldsymbol{Y}, \boldsymbol{X}) - H(\boldsymbol{X}) \tag{6}$$

Accordingly, injecting noise to the **raw images** can be formulated as follows Li (2022):

$$\begin{cases} H(\mathcal{T}; \boldsymbol{X} + \epsilon) = -\sum_{\boldsymbol{Y} \in \mathcal{Y}} p(\boldsymbol{Y}|\boldsymbol{X} + \epsilon) \log p(\boldsymbol{Y}|\boldsymbol{X} + \epsilon) \\ H(\mathcal{T}; \boldsymbol{X}\epsilon) = -\sum_{\boldsymbol{Y} \in \mathcal{Y}} p(\boldsymbol{Y}|\boldsymbol{X}\epsilon) \log p(\boldsymbol{Y}|\boldsymbol{X}\epsilon) \end{cases} \tag{7}$$

where $\epsilon$ represents additive or multiplicative noise respectively.

Here, we extend the analysis to **embedding space**. Given a set of $k$ embeddings $\boldsymbol{Z} = \{Z_1, Z_2, ..., Z_k\}$ from feature extraction of the raw images $\boldsymbol{X} = \{X_1, X_2, ..., X_k\}$, the label set $\boldsymbol{Y} = \{Y_1, Y_2, ..., Y_k\}$ can be regarded as sampling from $\boldsymbol{Y} \sim D_{\mathcal{Y}|\mathcal{Z}}$. The complexity of $\mathcal{T}$ on embeddings $\boldsymbol{Z}$ is:

$$H(\mathcal{T}; \boldsymbol{Z}) := H(\boldsymbol{Y}, \boldsymbol{Z}) - H(\boldsymbol{Z}) \tag{8}$$

The operation of proactively injecting noise in the latent space can be defined as:

$$\begin{cases} H(\mathcal{T}; \boldsymbol{Z} + \epsilon) := H(\boldsymbol{Y}, \boldsymbol{Z} + \epsilon) - H(\boldsymbol{Z}) \\ H(\mathcal{T}; \boldsymbol{Z}\epsilon) := H(\boldsymbol{Y}, \boldsymbol{Z}\epsilon) - H(\boldsymbol{Z}) \end{cases} \tag{9}$$

where $\epsilon$ represents additive or multiplicative noise respectively. The definition of Eq. 8 differs from (Li, 2022), as our method injects the noise into the latent representations instead of the raw images.

**Gaussian Noise** is one of the most common additive noises that appear in computer vision tasks. The Gaussian noise is independent and stochastic, obeying the Gaussian distribution $\epsilon \sim \mathcal{N}(\mu, \sigma^2)$. Injecting Gaussian noise into the embedding space, the complexity of the classification tasks is:

$$H(\mathcal{T}; \boldsymbol{Z} + \epsilon) = H(\boldsymbol{Y}, \boldsymbol{Z} + \epsilon) - H(\boldsymbol{Z}) \tag{10}$$

According to Eq. 4, the entropy change is formulated as:

$$
\begin{aligned}
\triangle S(\mathcal{T}, \boldsymbol{\epsilon}) =& H(\mathcal{T}; \boldsymbol{Z}) - H(\mathcal{T}; \boldsymbol{Z} + \boldsymbol{\epsilon}) \\
=& H(\boldsymbol{Y}, \boldsymbol{Z}) - H(\boldsymbol{Z}) - (H(\boldsymbol{Y}, \boldsymbol{Z} + \boldsymbol{\epsilon}) - H(\boldsymbol{Z})) \\
=& H(\boldsymbol{Y}, \boldsymbol{Z}) - H(\boldsymbol{Y}, \boldsymbol{Z} + \boldsymbol{\epsilon}) \\
=& \frac{1}{2} \log \frac{|\boldsymbol{\Sigma_Z}||\boldsymbol{\Sigma_Y} - \boldsymbol{\Sigma_{YZ}} \boldsymbol{\Sigma_Z^{-1}} \boldsymbol{\Sigma_{ZY}}|}{|\boldsymbol{\Sigma_{Z+\epsilon}}||\boldsymbol{\Sigma_Y} - \boldsymbol{\Sigma_{YZ}} \boldsymbol{\Sigma_{Z+\epsilon}^{-1}} \boldsymbol{\Sigma_{ZY}}|} \\
=& \frac{1}{2} \log \frac{1}{(1 + \sigma_\epsilon^2 \sum_{i=1}^k \frac{1}{\sigma_{Z_i}^2})(1 + \lambda \sum_{i=1}^k \frac{\text{cov}^2(Z_i, Y_i)}{\sigma_{X_i}^2 (\sigma_{Z_i}^2 \sigma_{Y_i}^2 - \text{cov}^2(Z_i, Y_i))})}
\end{aligned}
\tag{11}
$$

where $\lambda = \frac{\sigma_\epsilon^2}{1 + \sum_{i=1}^k \frac{1}{\sigma_{Z_i}^2}}$, $\sigma_\epsilon^2$ is the variance of the Gaussian noise, $\text{cov}(Z_i, Y_i)$ is the covariance of sample pair $(Z_i, Y_i)$, $\sigma_{Z_i}^2$ and $\sigma_{Y_i}^2$ are the variance of embedding $Z_i$ and label $Y_i$, respectively. We use the symbol $M$ to compare the quantity between the numerator and denominator of the logarithmic term. If $M$ is greater than 0, then the entropy change is greater than 0, and vice versa.

$$
\begin{aligned}
M =& 1 - (1 + \sigma_\epsilon^2 \sum_{i=1}^k \frac{1}{\sigma_{Z_i}^2})(1 + \lambda \sum_{i=1}^k \frac{\text{cov}^2(Z_i, Y_i)}{\sigma_{Z_i}^2 (\sigma_{Z_i}^2 \sigma_{Y_i}^2 - \text{cov}^2(Z_i, Y_i))}) \\
=& -\sigma_\epsilon^2 \sum_{i=1}^k \frac{1}{\sigma_{Z_i}^2} - \sigma_\epsilon^2 \sum_{i=1}^k \frac{1}{\sigma_{Z_i}^2} \cdot \lambda \sum_{i=1}^k \frac{\text{cov}^2(Z_i, Y_i)}{\sigma_{Z_i}^2 (\sigma_{Z_i}^2 \sigma_{Y_i}^2 - \text{cov}^2(Z_i, Y_i))} \\
& - \lambda \sum_{i=1}^k \frac{\text{cov}^2(Z_i, Y_i)}{\sigma_{Z_i}^2 (\sigma_{Z_i}^2 \sigma_{Y_i}^2 - \text{cov}^2(Z_i, Y_i))}
\end{aligned}
\tag{12}
$$

Since $\sigma_\epsilon^2 \geq 0$ and $\lambda \geq 0$, $\sigma_{Z_i}^2 \sigma_{Y_i}^2 - \text{cov}^2(Z_i, Y_i) = \sigma_{Z_i}^2 \sigma_{Y_i}^2 (1 - \rho_{Z_i Y_i}^2) \geq 0$, where $\rho_{Z_i Y_i}$ is the correlation coefficient between the embedding $Z_i$ and the corresponding label $Y_i$, **the sign of $M$ is negative**. Consequently, we conclude that **the injection of Gaussian noise into the embedding space is harmful to the task**. Detailed derivations can be found in App sec. B.

**Salt-and-pepper Noise** is a common multiplicative noise for images, causing unnatural changes such as black pixels in bright areas or white pixels in dark areas. Injecting salt-and-pepper noise into the embeddings, the entropy change can be formulated as:

$$
\begin{aligned}
\triangle S(\mathcal{T}, \boldsymbol{\epsilon}) =& H(\mathcal{T}; \boldsymbol{Z}) - H(\mathcal{T}; \boldsymbol{Z}\boldsymbol{\epsilon}) \\
=& H(\boldsymbol{Y}, \boldsymbol{Z}) - H(\boldsymbol{Z}) - (H(\boldsymbol{Y}, \boldsymbol{Z}\boldsymbol{\epsilon}) - H(\boldsymbol{Z})) \\
=& H(\boldsymbol{Y}, \boldsymbol{Z}) - H(\boldsymbol{Y}, \boldsymbol{Z}\boldsymbol{\epsilon}) \\
=& -\sum_{\boldsymbol{Z} \in \mathcal{Z}} \sum_{\boldsymbol{Y} \in \mathcal{Y}} p(\boldsymbol{Z}, \boldsymbol{Y}) \log p(\boldsymbol{Z}, \boldsymbol{Y}) + \sum_{\boldsymbol{Z} \in \mathcal{Z}} \sum_{\boldsymbol{Y} \in \mathcal{Y}} \sum_{\boldsymbol{\epsilon} \in \mathcal{E}} p(\boldsymbol{Z}\boldsymbol{\epsilon}, \boldsymbol{Y}) \log p(\boldsymbol{Z}\boldsymbol{\epsilon}, \boldsymbol{Y}) \\
=& \mathbb{E}\left[\log \frac{1}{p(\boldsymbol{Z}, \boldsymbol{Y})}\right] - \mathbb{E}\left[\log \frac{1}{p(\boldsymbol{Z}\boldsymbol{\epsilon}, \boldsymbol{Y})}\right] \\
=& \mathbb{E}\left[\log \frac{1}{p(\boldsymbol{Z}, \boldsymbol{Y})}\right] - \mathbb{E}\left[\log \frac{1}{p(\boldsymbol{Z}, \boldsymbol{Y})}\right] - \mathbb{E}\left[\log \frac{1}{p(\boldsymbol{\epsilon})}\right] \\
=& -H(\boldsymbol{\epsilon})
\end{aligned}
\tag{13}
$$

The negative entropy change indicates an increase in task complexity, thus we conclude that **salt-and-pepper noise is harmful noise**. Further details can be found in App sec. D.

**Linear Transform Noise** is obtained by applying an elementary transformation to the embeddings matrix, i.e., $\boldsymbol{\epsilon} = Q\boldsymbol{Z}$, where $Q$ is a linear transformation matrix. We name the $Q$ the quality matrix since it dictates whether the linear transform noise $\boldsymbol{\epsilon}$ will be positive or harmful. For the linear transform noise injection into the embeddings, the complexity of the task is formulated as:

$$
H(\mathcal{T}; \boldsymbol{Z} + Q\boldsymbol{Z}) = H(\boldsymbol{Y}; \boldsymbol{Z} + Q\boldsymbol{Z}) - H(\boldsymbol{Z})
\tag{14}
$$

The entropy change is then formulated as:

$$
\begin{aligned}
\triangle S(\mathcal{T}, Q\boldsymbol{Z}) =& H(\mathcal{T}; \boldsymbol{Z}) - H(\mathcal{T}; \boldsymbol{Z} + Q\boldsymbol{Z}) \\
=& H(\boldsymbol{Y}, \boldsymbol{Z}) - H(\boldsymbol{Z}) - (H(\boldsymbol{Y}, \boldsymbol{Z} + Q\boldsymbol{Z}) - H(\boldsymbol{Z})) \\
=& H(\boldsymbol{Y}, \boldsymbol{Z}) - H(\boldsymbol{Y}, \boldsymbol{Z} + Q\boldsymbol{Z}) \\
=& \frac{1}{2} \log \frac{|\boldsymbol{\Sigma_Z}||\boldsymbol{\Sigma_Y} - \boldsymbol{\Sigma_{YZ}}\boldsymbol{\Sigma_Z^{-1}}\boldsymbol{\Sigma_{ZY}}|}{|\boldsymbol{\Sigma_{(I+Q)Z}}||\boldsymbol{\Sigma_Y} - \boldsymbol{\Sigma_{YZ}}\boldsymbol{\Sigma_Z^{-1}}\boldsymbol{\Sigma_{XY}}|} \\
=& \frac{1}{2} \log \frac{1}{|I+Q|^2} \\
=& -\log|I+Q|
\end{aligned}
\tag{15}
$$

Linear transform noise can be made positive by formulating Eq. 15 as an optimization problem:

$$
\begin{aligned}
&\max_Q \triangle S(\mathcal{T}, Q\boldsymbol{Z}) \\
&s.t.\ rank(I+Q) = k \\
&\quad\quad [I+Q]_{ii} \geq [I+Q]_{ij}, i \neq j \\
&\quad\quad \|[I+Q]_i\|_1 = 1
\end{aligned}
\tag{16}
$$

The most important step is to ensure that $I + Q$ is full rank. The second constraint is to ensure the diagonal elements of matrix $(I + Q)$ are always larger than other elements of the same row, which helps make sure that the original information from that instance predominantly informs the prediction on an instance. Otherwise, the classifier might not be able to make accurate predictions. The third constraint is to maintain the norm of latent representations. Further details can be found in App sec. C. Thus **linear transform noise can be made positive noise with proper construction.**

## 3.2 NoisyNN

Building upon the theoretical analysis, we introduce NoisyNN, wherein the embeddings are injected with **positive linear transformation noises**. For a deep neural network, such as CNN or ViT, we choose an intermediate layer $l$ and inject linear transform noise to the embeddings $\boldsymbol{Z}$ under the constraints specified in Eq 16. In fact, many possible quality $Q$ matrices could satisfy these constraints, forming a design space. Here, we adopt a simple concrete construction of $Q$ that we call a *circular shift* as a working example, where each original $Z_i$ is perturbed by its neighbor $Z_{i+1}$.

We can formally express the circular shift noise injection strategy as follows: Let the scalar hyperparameter $\alpha \in [0, 1]$ define the perturbation strength. The quality matrix of *circular shift* $Q$ is implemented as $Q = \alpha * U - \alpha * I$, where $U_{i,j} = \delta_{i+1,j}$ with $\delta_{i+1,j}$ representing the Kronecker delta indicator Frankel (2011), and employing wrap-around (or "circular") indexing.

$$
Q = \begin{bmatrix}
-\alpha & \alpha & 0 & 0 & 0 \\
0 & -\alpha & \alpha & 0 & 0 \\
0 & 0 & -\alpha & \ddots & 0 \\
0 & 0 & 0 & \ddots & \alpha \\
\alpha & 0 & 0 & 0 & -\alpha
\end{bmatrix}
\tag{17}
$$

## 4 EXPERIMENTS

We conduct extensive experiments to assess the impact of various noises on classification tasks. Our experiments consider both CNNs and ViTs, across a wide range of model sizes, including ResNet-18, ResNet-34, ResNet-50, and ResNet-101 for the ResNet, and ViT-Tiny (ViT-T), ViT-Small (ViT-S), ViT-Base (ViT-B), and ViT-Large (ViT-L) for ViT. We show that these deep models benefit from positive noise. Detailed model specifications are in App E. By default, noise is injected into the **last layer embeddings** of these models and used in both the training and inference stages. Results with noise injection at different layers are in Ablation section 5. **While this work primarily focuses on image classification and domain adaptation, we additionally explored other related**

Table 1: ResNet with different kinds of noise on ImageNet. Vanilla means the vanilla model without noise. Accuracy is shown in percentage. Gaussian noise used here is subjected to standard normal distribution. In this table, NoisyNN refers to ResNet injected with linear transform noise, where the employed linear transform noise is derived in Eq. 17. The difference is shown in the bracket.

| Model | ResNet-18 | ResNet-34 | ResNet-50 | ResNet-101 |
|---|---|---|---|---|
| Vanilla | 69.10 (+0.00) | 73.27 (+0.00) | 75.90 (+0.00) | 77.84 (+0.00) |
| + Gaussian Noise | 67.55 (-1.55) | 71.87 (-1.40) | 75.57 (-0.33) | 77.28 (-0.56) |
| + Salt-and-pepper Noise | 60.65 (-8.45) | 69.83 (-3.44) | 51.79 (-24.11) | 60.14 (-17.70) |
| NoisyNN (ResNet-based) | **79.62 (+10.52)** | **80.05 (+6.78)** | **81.32 (+5.42)** | **81.91 (+4.07)** |

**tasks: Domain Generalization (App F.9), Text Classification (App F.10) and Object Detection (App F.11) to assess broader applicability of NoisyNN.**

**Experiment Setting.** The positive noise used in NoisyNN is generated via the formulation in Eq. 17. The Gaussian noise is generated from a normal distribution with zero mean and unit variance:

$$\epsilon \sim \mathcal{N}(0, 1) \tag{18}$$

For salt-and-pepper noise, we use the parameter $\beta$ to control the emergence probability:

$$\begin{cases} max(Z) & \text{if } p < \beta/2 \\ min(Z) & \text{if } p > 1 - \beta/2 \end{cases} \tag{19}$$

where $p$ is a probability generated by a random seed, $\beta \in [0, 1)$, and $Z$ is the embedding of an image.

More hyperparameter and training details are in App sec. E. To better see the effect of noise injection, we refrain from using other data augmentation by default. Later experiments compare NoisyNN with other data augmentation techniques (Table 18) and investigate the combination of them (Table 17).

## 4.1 IMAGE CLASSIFICATION RESULTS

We conduct extensive experiments on various image classification benchmarks. Here we mainly present results on large-scale ImageNet dataset Deng et al. (2009). Additional results on Tiny-ImageNet (Le & Yang, 2015), ImageNetV2 (Recht et al., 2019), ImageNet-A (Hendrycks et al., 2021), ImageNet-C Hendrycks & Dietterich (2019), CIFAR-10 (Krizhevsky et al., 2009), CIFAR-100 (Krizhevsky et al., 2009) and medical imaging dataset INbreast (Moreira et al., 2012) can be found in App Table 12, 13, 9, 10, 11 and 16. Note that NoisyNN does not incur additional computation costs beyond a simple linear transformation in the embedding space, runtime comparison with vanilla ViT can be found in App Table 24.

**CNN Family.** The experiment results of ResNets with different noises on the ImageNet dataset are summarized in Table 1. Our NoisyNN (ResNet-based) improves the classification accuracy by a large margin. While Gaussian and salt-and-pepper noise, which are theoretically proven to be harmful, degrades the performance. The results confirm our analysis in sec 3.1 and show that positive noise can effectively improve the image classification accuracy of CNN models.

**ViT Family.** The results of ViT with different noises on ImageNet are shown in Table 2. We can see that our NoisyNN (ViT-based) improves classification accuracy often by a large margin compared to vanilla ViT (e.g., **more than 5%** on ViT-S and ViT-B), while other noises degrade performances (even with extensive hyperparameter search, see App Table 14, 15). This again supports our theoretical analysis. In Table 3, we further compare NoisyNN with other prior works, such as DeiT Touvron et al. (2021), SwinTransformer Liu et al. (2021), DaViT Ding et al. (2022), and MaxViT Tu et al. (2022). NoisyNN has a significant advantage and achieves the new state-of-the-art result. Note that JFT-300M and JFT-4B datasets are private and not publicly available Sun et al. (2017).

**Deriving Optimal Quality Matrix.** A key advantage of our framework is the ability to analytically derive the optimal quality matrix $Q$, compared to many other data augmentation methods that need to search over large hyperparameter space or need domain knowledge for ad-hoc design (Cubuk et al., 2020).

As depicted in Equation 16, it is intriguing to explore the optimal quality matrix $Q$ that maximizes the entropy change while adhering to the constraints. This optimization task is equivalent to minimizing

Table 2: ViT with different kinds of noise on ImageNet. Vanilla means the vanilla model without injecting noise. Accuracy is shown in percentage. Gaussian noise used here is subjected to standard normal distribution. In this table, NoisyNN refers to ViT injected with linear transform noise, where the employed linear transform noise is derived in Eq. 17. The difference is shown in the bracket. Note **without additional data, ViT-L exhibits overfitting on ImageNet** Dosovitskiy et al. (2020) Steiner et al. (2021).

| Model | ViT-T | ViT-S | ViT-B | ViT-L |
|---|---|---|---|---|
| Vanilla | 79.34 (+0.00) | 81.88 (+0.00) | 84.33 (+0.00) | 88.64 (+0.00) |
| + Gaussian Noise | 79.10 (-0.24) | 81.80 (-0.08) | 83.41 (-0.92) | 85.92 (-2.72) |
| + Salt-and-pepper Noise | 78.64 (-0.70) | 81.75 (-0.13) | 82.40 (-1.93) | 85.15 (-3.49) |
| NoisyNN (ViT-based) | **80.69 (+1.35)** | **87.27 (+5.39)** | **89.99 (+5.66)** | **88.97 (+0.33)** |

Table 3: Comparison between NoisyNN with other ViT variants. Showing Top-1 Accuracy (%) and standard deviation. Values for other methods are copied from original papers, some of which did not report standard deviation. Circular Shift Q is referred to Eq. 17. Optimal Q is **analytically derived** in Eq. 20. The best performance is marked in bold black.

| Model | Top1 Acc. | Params. | Image Res. | Pretrained Dataset |
|---|---|---|---|---|
| ViT-B Dosovitskiy et al. (2020) | 84.3 | 86M | $224 \times 224$ | ImageNet 21k |
| DeiT-B Touvron et al. (2021) | 85.7 | 86M | $224 \times 224$ | ImageNet 21k |
| SwinTransformer-B Liu et al. (2021) | 86.4 | 88M | $384 \times 384$ | ImageNet 21k |
| DaViT-B Ding et al. (2022) | 86.9 | 88M | $384 \times 384$ | ImageNet 21k |
| MaxViT-B Tu et al. (2022) | 88.8 | 119M | $512 \times 512$ | JFT-300M (Private) |
| ViT-22B Dehghani et al. (2023) | 89.5 | 21743M | $224 \times 224$ | JFT-4B (Private) |
| NoisyNN (ViT-based, Circular Shift Q) | **89.9±0.5** | 86M | $224 \times 224$ | ImageNet 21k |
| NoisyNN (ViT-based, Circular Shift Q) | **91.3±0.4** | 86M | $384 \times 384$ | ImageNet 21k |
| NoisyNN (ViT-based, Optimal Q) | **93.1±0.9** | 86M | $224 \times 224$ | ImageNet 21k |
| NoisyNN (ViT-based, Optimal Q) | **94.8±1.1** | 86M | $384 \times 384$ | ImageNet 21k |

the determinant of the matrix sum of $I$ and $Q$. Here, we directly present the **analytically derived** optimal quality matrix $Q$:

$$Q_{optimal} = \text{diag}\left(\frac{1}{k+1} - 1, \ldots, \frac{1}{k+1} - 1\right) + \frac{1}{k+1}\mathbf{1}_{k \times k} \tag{20}$$

where $k$ is the training data size, and $\mathbf{1}_{k \times k}$ is a matrix of ones. The corresponding upper bound of the entropy change is:

$$\triangle S(\mathcal{T}, Q_{optimal}\mathbf{Z}) = (k-1)\log(k+1) \tag{21}$$

Detailed derivations are provided in the App C.1.1. We find that the upper bound of the entropy change of injecting positive noise is determined by the number of data samples, i.e., the scale of the dataset. The larger the dataset, the more pronounced the effect of injecting positive noise into the embeddings.

## 4.2 DOMAIN ADAPTATION RESULTS

Unsupervised domain adaptation (UDA) aims to learn transferable knowledge across the source and target domains with different distributions Pan & Yang (2009) Wei et al. (2018). Recently, transformer-based methods achieved the state-of-the-art (SOTA) results on UDA. Here, we evaluate NoisyNN on the widely used UDA benchmarks, including the Office Home dataset Venkateswara et al. (2017) and the VisDA2017 dataset Peng et al. (2017). The positive noise is generated via Eq. 17, and injected into the last layer embeddings of the models, same as sec. 4.1. More details on the datasets and experiment settings are in App sec. G. We use the same objective function as TVT Yang et al. (2023a), which is the first work that adopts Transformer-based architecture for UDA. The results are shown in Table 4 and 5. Our NoisyNN (TVT-based) achieves SOTA on VisDA2017 and is competitive on Office-Home. These results demonstrate that positive noise also works in the domain adaptation tasks, where out-of-distribution (OOD) data exists.

Table 4: Comparison with SOTA methods on **Office-Home**. Above the middle black line are methods based on CNNs, while below the middle black line are methods based on ViTs. The best performance is marked in bold black.

| Method | Ar→Cl | Ar→Pr | Ar→Re | Cl→Ar | Cl→Pr | Cl→Re | Pr→Ar | Pr→Cl | Pr→Re | Re→Ar | Re→Cl | Re→Pr | Avg. |
|---|---|---|---|---|---|---|---|---|---|---|---|---|---|
| ResNet-50He et al. (2016) | 44.9 | 66.3 | 74.3 | 51.8 | 61.9 | 63.6 | 52.4 | 39.1 | 71.2 | 63.8 | 45.9 | 77.2 | 59.4 |
| MinEntGrandvalet & Bengio (2004) | 51.0 | 71.9 | 77.1 | 61.2 | 69.1 | 70.1 | 59.3 | 48.7 | 77.0 | 70.4 | 53.0 | 81.0 | 65.8 |
| SAFNXu et al. (2019) | 52.0 | 71.7 | 76.3 | 64.2 | 69.9 | 71.9 | 63.7 | 51.4 | 77.1 | 70.9 | 57.1 | 81.5 | 67.3 |
| CDAN+ELong et al. (2018) | 54.6 | 74.1 | 78.1 | 63.0 | 72.2 | 74.1 | 61.6 | 52.3 | 79.1 | 72.3 | 57.3 | 82.8 | 68.5 |
| DCANLi et al. (2020) | 54.5 | 75.7 | 81.2 | 67.4 | 74.0 | 76.3 | 67.4 | 52.7 | 80.6 | 74.1 | 59.1 | 83.5 | 70.5 |
| BNM Cui et al. (2020) | 56.7 | 77.5 | 81.0 | 67.3 | 76.3 | 77.1 | 65.3 | 55.1 | 82.0 | 73.6 | 57.0 | 84.3 | 71.1 |
| SHOTLiang et al. (2020) | 57.1 | 78.1 | 81.5 | 68.0 | 78.2 | 78.1 | 67.4 | 54.9 | 82.2 | 73.3 | 58.8 | 84.3 | 71.8 |
| ATDOC-NALiang et al. (2021) | 58.3 | 78.8 | 82.3 | 69.4 | 78.2 | 78.2 | 67.1 | 56.0 | 82.7 | 72.0 | 58.2 | 85.5 | 72.2 |
| ViT-BDosovitskiy et al. (2020) | 54.7 | 83.0 | 87.2 | 77.3 | 83.4 | 85.6 | 74.4 | 50.9 | 87.2 | 79.6 | 54.8 | 88.8 | 75.5 |
| TVT-BYang et al. (2023a) | 74.9 | 86.8 | 89.5 | 82.8 | 88.0 | 88.3 | 79.8 | 71.9 | 90.1 | 85.5 | 74.6 | 90.6 | 83.6 |
| CDTrans-BXu et al. (2022) | 68.8 | 85.0 | 86.9 | 81.5 | 87.1 | 87.3 | 79.6 | 63.3 | 88.2 | 82.0 | 66.0 | 90.6 | 80.5 |
| SSRT-B Sun et al. (2022) | 75.2 | 89.0 | 91.1 | 85.1 | 88.3 | 90.0 | 85.0 | 74.2 | 91.3 | 85.7 | 78.6 | 91.8 | 85.4 |
| NoisyNN (TVT-based) | **78.3** | **90.6** | **91.9** | **87.8** | **92.1** | **91.9** | **85.8** | **78.7** | **93.0** | **88.6** | **80.6** | **93.5** | **87.7** |

Table 5: Comparison with SOTA methods on **Visda2017**. Above the middle line are methods based on CNNs, while below the middle line are methods based on ViTs. The best performance is marked in bold.

| Method | plane | bcycl | bus | car | horse | knife | mcycl | person | plant | sktbrd | train | truck | Avg. |
|---|---|---|---|---|---|---|---|---|---|---|---|---|---|
| ResNet-50He et al. (2016) | 55.1 | 53.3 | 61.9 | 59.1 | 80.6 | 17.9 | 79.7 | 31.2 | 81.0 | 26.5 | 73.5 | 8.5 | 52.4 |
| DANNGanin & Lempitsky (2015) | 81.9 | 77.7 | 82.8 | 44.3 | 81.2 | 29.5 | 65.1 | 28.6 | 51.9 | 54.6 | 82.8 | 7.8 | 57.4 |
| MinEntGrandvalet & Bengio (2004) | 80.3 | 75.5 | 75.8 | 48.3 | 77.9 | 27.3 | 69.7 | 40.2 | 46.5 | 46.6 | 79.3 | 16.0 | 57.0 |
| SAFNXu et al. (2019) | 93.6 | 61.3 | 84.1 | 70.6 | 94.1 | 79.0 | 91.8 | 79.6 | 89.9 | 55.6 | 89.0 | 24.4 | 76.1 |
| CDAN+ELong et al. (2018) | 85.2 | 66.9 | 83.0 | 50.8 | 84.2 | 74.9 | 88.1 | 74.5 | 83.4 | 76.0 | 81.9 | 38.0 | 73.9 |
| BNM Cui et al. (2020) | 89.6 | 61.5 | 76.9 | 55.0 | 89.3 | 69.1 | 81.3 | 65.5 | 90.0 | 47.3 | 89.1 | 30.1 | 70.4 |
| CGDMDu et al. (2021) | 93.7 | 82.7 | 73.2 | 68.4 | 92.9 | 94.5 | 88.7 | 82.1 | 93.4 | 82.5 | 86.8 | 49.2 | 82.3 |
| SHOTLiang et al. (2020) | 94.3 | 88.5 | 80.1 | 57.3 | 93.1 | 93.1 | 80.7 | 80.3 | 91.5 | 89.1 | 86.3 | 58.2 | 82.9 |
| ViT-BDosovitskiy et al. (2020) | 97.7 | 48.1 | 86.6 | 61.6 | 78.1 | 63.4 | 94.7 | 10.3 | 87.7 | 47.7 | 94.4 | 35.5 | 67.1 |
| TVT-BYang et al. (2023a) | 92.9 | 85.6 | 77.5 | 60.5 | 93.6 | 98.2 | 89.4 | 76.4 | 93.6 | 92.0 | 91.7 | 55.7 | 83.9 |
| CDTrans-BXu et al. (2022) | 97.1 | 90.5 | 82.4 | 77.5 | 96.6 | 96.1 | 93.6 | **88.6** | **97.9** | 86.9 | 90.3 | 62.8 | 88.4 |
| SSRT-B Sun et al. (2022) | **98.9** | 87.6 | **89.1** | **84.8** | 98.3 | **98.7** | **96.3** | 81.1 | 94.9 | 97.9 | 94.5 | 43.1 | 88.8 |
| NoisyNN (TVT-based) | 98.8 | **95.5** | 84.8 | 73.7 | **98.5** | 97.2 | 95.1 | 76.5 | 95.9 | **98.4** | **98.3** | **67.2** | **90.0** |

## 5 ABLATION

**Design Choice.** We conduct a comprehensive ablation on the two critical design choices of NoisyNN: the perturbation strength $\alpha$ and the layer $l$ where the noise is injected. Results are shown in Fig. 2. We observe that injecting positive noise into deeper layers often yields better performance. Furthermore, within the region $\alpha < 0.5$ (the constraint in Eq. 16), a larger $\alpha$ provides better performance, which aligns with theoretical analysis, as a larger $\alpha$ induces a more substantial entropy change (Eq. 15, 17).

Table 6: Variants of ViT with different kinds of noise on TinyImageNet. Vanilla means the vanilla model without noise. Accuracy is shown in percentage. Gaussian noise used here is subjected to standard normal distribution. Linear transform noise used in this table is designed to be positive noise. The difference is shown in the bracket.

| Model | DeiT | SwinTransformer | BeiT | ConViT |
|---|---|---|---|---|
| Vanilla | 85.02 (+0.00) | 90.84 (+0.00) | 88.64 (+0.00) | 90.69 (+0.00) |
| + Gaussian Noise | 84.70 (-0.32) | 90.34 (-0.50) | 88.40 (-0.24) | 90.40 (-0.29) |
| + Salt-and-pepper Noise | 84.03 (-1.01) | 87.12 (-3.72) | 42.18 (-46.46) | 89.93 (-0.76) |
| + Linear Transform Noise | **86.50 (+1.48)** | **95.68 (+4.84)** | **91.78 (+3.14)** | **93.07 (+2.38)** |
| Params. | 86M | 87M | 86M | 86M |

**Compatibility with Other Architectures.** We also proactively inject noise into other ViT variants, such as DeiT Touvron et al. (2021), Swin Transformer Liu et al. (2021), BEiT Bao et al. (2021), and ConViT d'Ascoli et al. (2021). The results are reported in Table 6. As expected, these variants of ViTs benefit from the positive noise. These additional four ViT variants are at the base scale, whose parameters are listed in the table's last row. For a fair comparison, we use identical experimental

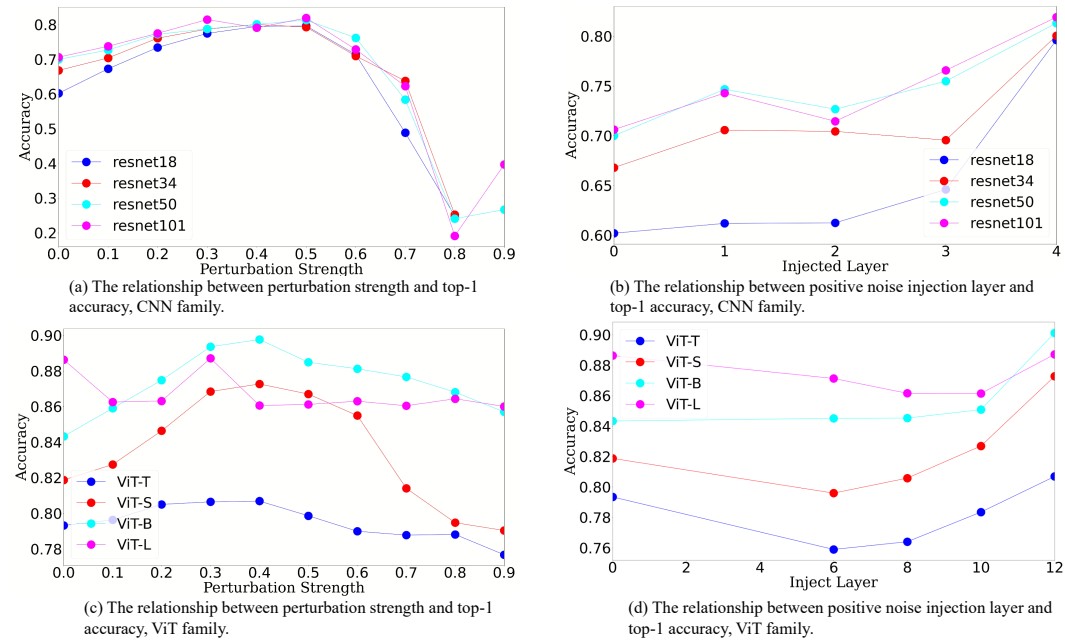

Figure 2: Ablation on perturbation strength (a, c) and noise injection layer (b, d). Showing top-1 accuracy on ImageNet. The positive noise refers to the linear transform noise from 17. For parts (a) and (c), the linear transform positive noise is injected into the last layer. Note that in (d) ViT-L has 24 layers while the other variants have 12. For visualization purpose we show the performance up to layer 12.

settings for each kind of experiment. For example, we use the identical setting for vanilla ConViT, ConViT with different kinds of noise. From the experimental results, we can observe that the different variants of ViT significantly improve prediction accuracy through injecting positive noise. The results on different scale datasets and variants of the ViT family demonstrate that positive noise can universally improve the model performance.

**Comparison with common data augmentation techniques.** To compare NoisyNN with data augmentation techniques and explore whether our proposed NoisyNN is compatible with existing data augmentation techniques, we conduct corresponding experiments in App Table 18. The results demonstrate that linear transform positive noise significantly outperforms the common data augmentation techniques evaluated. Integrating linear transform positive noise with other common data augmentation techniques does not substantially change performance.

## 6   CONCLUSION AND LIMITATION

This study theoretically and empirically explores the impacts of injecting noise into the embedding space of deep neural networks. We show that Gaussian and salt-and-pepper noise are harmful noises while linear transform noise can be made positive noise under proper construction and thus positively affect deep neural networks. The results of the extensive experiments on the 15 datasets, which include datasets with significant domain shifts, demonstrate the efficacy of our approach. Our study provides the community with a new paradigm for improving model performance. However, the theoretical analysis of the current study is tailored to classification tasks. While we preliminarily explored the applicability of the NoisyNN framework for other tasks (Domain Generalization, Text Classification, and Object Detection), more study is needed to confirm its effectiveness for those tasks, which might entail conducting theoretical analyses and extensive empirical evaluations.

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

# Supplementary Material

In this supplement, we provide:

## A    THEORETICAL FOUNDATIONS OF TASK ENTROPY

This section provides the theoretical foundations of task entropy, quantifying the complexity of learning tasks. The concept of task entropy was first proposed for the image level and formulated as Li (2022):

$$H(\mathcal{T}; \boldsymbol{X}) = - \sum_{\boldsymbol{Y} \in \mathcal{Y}} p(\boldsymbol{Y}|\boldsymbol{X}) \log p(\boldsymbol{Y}|\boldsymbol{X}) \tag{22}$$

The image $\boldsymbol{X}$ in the dataset are supposed to be independent of each other, as are the labels $\boldsymbol{Y}$. However, $\boldsymbol{X}$ and $\boldsymbol{Y}$ are not independent because of the correlation between a data sample $X$ and its corresponding label $Y$. Essentially, the task entropy is the entropy of $p(Y|X)$. Following the principle of task entropy, compelling evidence suggests that diminishing task complexity via reducing information entropy can enhance overall model performance Li (2022); Jain et al. (2023a); Zhang et al. (2023).

Inspired by the concept of task entropy at the image level, we explore its extension to the latent space. The task entropy from the perspective of embeddings is defined as:

$$H(\mathcal{T}; \boldsymbol{Z}) := H(\boldsymbol{Y}, \boldsymbol{Z}) - H(\boldsymbol{Z}) \tag{23}$$

where $\boldsymbol{Z}$ are the embeddings of the images $\boldsymbol{X}$. Here, we assume that the embedding $\boldsymbol{Z}$ and the vectorized label $\boldsymbol{Y}$ follow a multivariate normal distribution. We can transform the unknown distributions of $\boldsymbol{Z}$ and $\boldsymbol{Y}$ to approximately conform to normality by utilizing existing techniques such as reparameterization tricks Kingma & Welling (2013); Van Den Oord & Vinyals (2017). After approximate transformation, the distribution of $\boldsymbol{Z}$ and $\boldsymbol{Y}$ can be expressed as:

$$\boldsymbol{Z} \sim \mathcal{N}(\boldsymbol{\mu_Z}, \boldsymbol{\Sigma_Z}), \boldsymbol{Y} \sim \mathcal{N}(\boldsymbol{\mu_Y}, \boldsymbol{\Sigma_Y}) \tag{24}$$

where

$$
\begin{aligned}
\boldsymbol{\mu_Z} &= \mathbb{E}[\boldsymbol{Z}] = (\mathbb{E}[Z_1], \mathbb{E}[Z_2], ..., \mathbb{E}[Z_k]])^T \\
\boldsymbol{\mu_Y} &= \mathbb{E}[\boldsymbol{Y}] = (\mathbb{E}[Y_1], \mathbb{E}[Y_2], ..., \mathbb{E}[Y_k]])^T \\
\boldsymbol{\Sigma_Z} &= \mathbb{E}[(\boldsymbol{Z} - \boldsymbol{\mu_Z})(\boldsymbol{Z} - \boldsymbol{\mu_Z})^T] \\
\boldsymbol{\Sigma_Y} &= \mathbb{E}[(\boldsymbol{Y} - \boldsymbol{\mu_Y})(\boldsymbol{Y} - \boldsymbol{\mu_Y})^T]
\end{aligned}
\tag{25}
$$

$k$ is the number of samples in the dataset, and $T$ represents the transpose of the matrix.

Then the conditional distribution of $\boldsymbol{Y}$ given $\boldsymbol{Z}$ is also normally distributed Mood (1950); Johnson et al. (1995), which can be formulated as:

$$\boldsymbol{Y}|\boldsymbol{Z} \sim \mathcal{N}(\mathbb{E}(\boldsymbol{Y}|\boldsymbol{Z} = Z), var(\boldsymbol{Y}|\boldsymbol{Z} = Z)) \tag{26}$$

where $\mathbb{E}(\boldsymbol{Y}|\boldsymbol{Z} = Z)$ is the mean of the label set $\boldsymbol{Y}$ given a sample $\boldsymbol{Z} = Z$ from the embeddings, and $var(\boldsymbol{Y}|\boldsymbol{Z} = Z)$ is the variance of $\boldsymbol{Y}$ given a sample from the embeddings. The conditional mean $\mathbb{E}[(\boldsymbol{Y}|\boldsymbol{Z} = Z)]$ and conditional variance $var(\boldsymbol{Y}|\boldsymbol{Z} = Z)$ can be calculated as:

$$\boldsymbol{\mu_{Y|Z=Z}} = \mathbb{E}[(\boldsymbol{Y}|\boldsymbol{Z} = Z)] = \boldsymbol{\mu_Y} + \boldsymbol{\Sigma_{YZ}}\boldsymbol{\Sigma_Z^{-1}}(Z - \boldsymbol{\mu_Z}) \tag{27}$$

$$\Sigma_{Y|Z=Z} = var(Y|Z=Z) = \Sigma_Y - \Sigma_{YX}\Sigma_Z^{-1}\Sigma_{ZY} \tag{28}$$

where $\Sigma_{YZ}$ and $\Sigma_{ZY}$ are the cross-covariance matrices between $Y$ and $Z$, and between $Z$ and $Y$, respectively, and $\Sigma_Z^{-1}$ denotes the inverse of the covariance matrix of $Z$.

Now, we shall obtain the task entropy:

$$
\begin{aligned}
H(\mathcal{T}; \boldsymbol{Z}) &= -\sum_{\boldsymbol{Y} \in \mathcal{Y}} p(\boldsymbol{Y}|\boldsymbol{Z}) \log p(\boldsymbol{Y}|\boldsymbol{Z}) \\
&= -\mathbb{E}[\log p(\boldsymbol{Y}|\boldsymbol{Z})] \\
&= -\mathbb{E}[\log[(2\pi)^{-k/2}|\boldsymbol{\Sigma_Z}|^{-1/2}\exp(-\frac{1}{2}(\boldsymbol{Y}|\boldsymbol{Z} - \boldsymbol{\mu_{Y|Z}})^T\boldsymbol{\Sigma_{Y|Z}^{-1}}(\boldsymbol{Y}|\boldsymbol{Z} - \boldsymbol{\mu_{Y|Z}}))]] \\
&= \frac{k}{2}(1 + \log(2\pi)) + \frac{1}{2}\log|\boldsymbol{\Sigma_{Y|Z}}|
\end{aligned}
\tag{29}
$$

Therefore, for a specific set of embeddings, we can find that the task entropy is only related to the variance of the $Y|Z$.

As we proactively inject different kinds of noises into the latent space, the task entropy with noise injection is defined as :

$$
\begin{cases}
H(\mathcal{T}; \boldsymbol{Z} + \boldsymbol{\epsilon}) := H(\boldsymbol{Y}; \boldsymbol{Z} + \boldsymbol{\epsilon}) - H(\boldsymbol{Z}) & \boldsymbol{\epsilon} \text{ is additive noise} \\
H(\mathcal{T}; \boldsymbol{Z}\boldsymbol{\epsilon}) := H(\boldsymbol{Y}; \boldsymbol{Z}\boldsymbol{\epsilon}) - H(\boldsymbol{Z}) & \boldsymbol{\epsilon} \text{ is multiplicative noise}
\end{cases}
\tag{30}
$$

Equation 30 diverges from the conventional definition of conditional entropy as our method introduces noise into the latent representations instead of the original images. The noises examined in this study are classified into additive and multiplicative categories. In the subsequent sections, we analyze the changes in task entropy resulting from the injection of common noises into the embeddings.

## B THE IMPACT OF GAUSSIAN NOISE ON TASK ENTROPY

We begin by examining the impact of Gaussian noise on task entropy from the perspective of latent space.

### B.1 INJECT GAUSSIAN NOISE INTO EMBEDDINGS

In this case, the task complexity is formulated as:

$$H(\mathcal{T}; \boldsymbol{Z} + \boldsymbol{\epsilon}) = H(\boldsymbol{Y}; \boldsymbol{Z} + \boldsymbol{\epsilon}) - H(\boldsymbol{Z}). \tag{31}$$

Take advantage of the definition of task entropy, thus, the entropy change of injecting Gaussian noise in the latent space can be formulated as:

$$
\begin{aligned}
\triangle S(\mathcal{T}, \boldsymbol{\epsilon}) &= H(\mathcal{T}; \boldsymbol{Z}) - H(\mathcal{T}; \boldsymbol{Z} + \boldsymbol{\epsilon}) \\
&= H(\boldsymbol{Y}; \boldsymbol{Z}) - H(\boldsymbol{Z}) - (H(\boldsymbol{Y}; \boldsymbol{Z} + \boldsymbol{\epsilon}) - H(\boldsymbol{Z})) \\
&= H(\boldsymbol{Y}; \boldsymbol{Z}) - H(\boldsymbol{Y}; \boldsymbol{Z} + \boldsymbol{\epsilon}) \\
&= \frac{1}{2}\log|\boldsymbol{\Sigma_{Y|Z}}| + \frac{1}{2}\log|\boldsymbol{\Sigma_Z}| - \frac{1}{2}\log|\boldsymbol{\Sigma_{Y|Z+\epsilon}}| - \frac{1}{2}\log|\boldsymbol{\Sigma_{Z+\epsilon}}| \\
&= \frac{1}{2}\log\frac{|\boldsymbol{\Sigma_Z}||\boldsymbol{\Sigma_{Y|Z}}|}{|\boldsymbol{\Sigma_{Z+\epsilon}}||\boldsymbol{\Sigma_{Y|Z+\epsilon}}|} \\
&= \frac{1}{2}\log\frac{|\boldsymbol{\Sigma_Z}||\boldsymbol{\Sigma_Y} - \boldsymbol{\Sigma_{YZ}}\boldsymbol{\Sigma_Z^{-1}}\boldsymbol{\Sigma_{ZY}}|}{|\boldsymbol{\Sigma_{Z+\epsilon}}||\boldsymbol{\Sigma_Y} - \boldsymbol{\Sigma_{YZ}}\boldsymbol{\Sigma_{Z+\epsilon}^{-1}}\boldsymbol{\Sigma_{ZY}}|}
\end{aligned}
\tag{32}
$$

where $\Sigma_{Y|Z+\epsilon} = \Sigma_Y - \Sigma_{Y(Z+\epsilon)}\Sigma_{Z+\epsilon}^{-1}\Sigma_{(Z+\epsilon)Y}$. Since the Gaussian noise is independent of $Z$ and $Y$, we have $\Sigma_{Y(Z+\epsilon)} = \Sigma_{(Z+\epsilon)Y} = \Sigma_{YZ}$. The corresponding proof is:

$$
\begin{aligned}
\Sigma_{(Z+\epsilon)Y} =& \mathbb{E}\left[(Z+\epsilon) - \mu_{Z+\epsilon}\right]\mathbb{E}\left[Y - \mu_Y\right] \\
=& \mathbb{E}\left[(Z+\epsilon)Y\right] - \mu_Y\mathbb{E}\left[(Z+\epsilon)\right] - \mu_{Z+\epsilon}\mathbb{E}\left[Y\right] + \mu_Y\mu_{Z+\epsilon} \\
=& \mathbb{E}\left[(Z+\epsilon)Y\right] - \mu_Y\mathbb{E}\left[(Z+\epsilon)\right] \\
=& \mathbb{E}\left[ZY\right] + \mathbb{E}\left[\epsilon Y\right] - \mu_Y\mu_Z - \mu_Y\mu_\epsilon \\
=& \mathbb{E}\left[ZY\right] - \mu_Y\mu_Z \\
=& \Sigma_{ZY}
\end{aligned}
\tag{33}
$$

Obviously,

$$
\begin{cases}
\triangle S(\mathcal{T}, \epsilon) > 0 & if \ \frac{|\Sigma_Z||\Sigma_{Y|Z}|}{|\Sigma_{Z+\epsilon}||\Sigma_{Y|Z+\epsilon}|} > 1 \\
\triangle S(\mathcal{T}, \epsilon) \le 0 & if \ \frac{|\Sigma_Z||\Sigma_{Y|Z}|}{|\Sigma_{Z+\epsilon}||\Sigma_{Y|Z+\epsilon}|} \le 1
\end{cases}
\tag{34}
$$

To find the relationship between $|\Sigma_Z||\Sigma_{Y|Z}|$ and $|\Sigma_{Z+\epsilon}||\Sigma_{Y|Z+\epsilon}|$, we need to determine the subterms in each of them. As we mentioned in the previous section, the embeddings of the images are independent of each other, and so are the labels.

$$
\begin{aligned}
\Sigma_Y =& \mathbb{E}[(Y - \mu_Y)(Y - \mu_Y)^T] \\
=& \mathbb{E}[YY^T] - \mu_Y\mu_Y^T \\
=& \text{diag}(\sigma_{Y_1}^2, ..., \sigma_{Y_k}^2)
\end{aligned}
\tag{35}
$$

where

$$
\begin{cases}
\mathbb{E}\left[Y_iY_j\right] - \mu_{Y_i}\mu_{Y_j} = 0, & i \ne j \\
\mathbb{E}\left[Y_iY_j\right] - \mu_{Y_i}\mu_{Y_j} = \sigma_{Y_i}^2, & i = j
\end{cases}
\tag{36}
$$

The same procedure can be applied to $\Sigma_{Y(Z+\epsilon)}$ and $\Sigma_{Z+\epsilon}$. Therefore, We can obtain that $\Sigma_Y = \text{diag}(\sigma_{Y_1}^2, ..., \sigma_{Y_k}^2)$,

$$
\Sigma_{Y(Z+\epsilon)} = \text{diag}(\text{cov}(Y_1, X_1 + \epsilon), ..., \text{cov}(Y_k, X_k + \epsilon))
\tag{37}
$$

and $\Sigma_{Z+\epsilon}$ is:

$$
\begin{aligned}
\Sigma_{Z+\epsilon} =& \begin{bmatrix}
\sigma_{Z_1}^2 + \sigma_\epsilon^2 & \sigma_\epsilon^2 & ... & \sigma_\epsilon^2 & \sigma_\epsilon^2 \\
\sigma_\epsilon^2 & \sigma_{Z_2}^2 + \sigma_\epsilon^2 & ... & \sigma_\epsilon^2 & \sigma_\epsilon^2 \\
\vdots & \vdots & & \vdots & \vdots \\
\sigma_\epsilon^2 & \sigma_\epsilon^2 & ... & \sigma_{Z_{k-1}}^2 + \sigma_\epsilon^2 & \sigma_\epsilon^2 \\
\sigma_\epsilon^2 & \sigma_\epsilon^2 & ... & \sigma_\epsilon^2 & \sigma_{Z_k}^2 + \sigma_\epsilon^2
\end{bmatrix} \\
=& \text{diag}(\sigma_{Z_1}^2, ..., \sigma_{Z_k}^2)I_k + \sigma_\epsilon^2 \mathbf{1}_k
\end{aligned}
\tag{38}
$$

where $I_k$ is a $k \times k$ identity matrix and $\mathbf{1}_k$ is a all ones $k \times k$ matrix. We use $U$ to represent $\text{diag}(\sigma_{Z_1}^2, ..., \sigma_{Z_k}^2)I_k$, and $u$ to represent a all ones vector $[1, ..., 1]^T$. Thanks to the Sherman–Morrison Formula Sherman & Morrison (1949) and Woodbury Formula Woodbury (1950), we can obtain the inverse of $\Sigma_{Z+\epsilon}$ as:

$$
\begin{aligned}
\Sigma_{Z+\epsilon}^{-1} =& (U + \sigma_\epsilon^2 uu^T)^{-1} \\
=& U^{-1} - \frac{\sigma_\epsilon^2}{1 + \sigma_\epsilon^2 u^T U^{-1}u}U^{-1}uu^T U^{-1} \\
=& U^{-1} - \frac{\sigma_\epsilon^2}{1 + \sum_{i=1}^k \frac{1}{\sigma_{Z_i}^2}}U^{-1}\mathbf{1}_k U^{-1} \\
=& \lambda \begin{bmatrix}
\frac{1}{\lambda\sigma_{Z_1}^2} - \frac{1}{\sigma_{Z_1}^4} & -\frac{1}{\sigma_{Z_1}^2\sigma_{Z_2}^2} & ... & -\frac{1}{\sigma_{Z_1}^2\sigma_{Z_{k-1}}^2} & -\frac{1}{\sigma_{Z_1}^2\sigma_{Z_k}^2} \\
-\frac{1}{\sigma_{Z_2}^2\sigma_{Z_1}^2} & \frac{1}{\lambda\sigma_{Z_2}^2} - \frac{1}{\sigma_{Z_2}^4} & ... & -\frac{1}{\sigma_{Z_2}^2\sigma_{Z_{k-1}}^2} & -\frac{1}{\sigma_{Z_2}^2\sigma_{Z_k}^2} \\
\vdots & \vdots & & \vdots & \vdots \\
-\frac{1}{\sigma_{Z_{k-1}}^2\sigma_{Z_1}^2} & -\frac{1}{\sigma_{Z_{k-1}}^2\sigma_{Z_2}^2} & ... & \frac{1}{\lambda\sigma_{Z_{k-1}}^2} - \frac{1}{\sigma_{Z_{k-1}}^4} & -\frac{1}{\sigma_{Z_{k-1}}^2\sigma_{Z_k}^2} \\
-\frac{1}{\sigma_{Z_k}^2\sigma_{Z_1}^2} & -\frac{1}{\sigma_{Z_k}^2\sigma_{Z_2}^2} & ... & -\frac{1}{\sigma_{Z_k}^2\sigma_{Z_{k-1}}^2} & \frac{1}{\lambda\sigma_{Z_k}^2} - \frac{1}{\sigma_{Z_k}^4}
\end{bmatrix}
\end{aligned}
\tag{39}
$$

where $\boldsymbol{U}^{-1} = \mathrm{diag}((\sigma_{Z_1}^2)^{-1}, ..., (\sigma_{Z_k}^2)^{-1})$ and $\lambda = \frac{\sigma_\epsilon^2}{1 + \sum_{i=1}^k \frac{1}{\sigma_{Z_i}^2}}$.

Therefore, substitute Equation 39 into $|\boldsymbol{\Sigma_Y} - \boldsymbol{\Sigma_{Y(Z+\epsilon)}} \boldsymbol{\Sigma_{Z+\epsilon}^{-1}} \boldsymbol{\Sigma_{(Z+\epsilon)Y}}|$, we can obtain:

$$
\begin{aligned}
&|\boldsymbol{\Sigma_Y} - \boldsymbol{\Sigma_{Y(Z+\epsilon)}} \boldsymbol{\Sigma_{Z+\epsilon}^{-1}} \boldsymbol{\Sigma_{(Z+\epsilon)Y}}| \\
&= \left| \begin{bmatrix} \sigma_{Y_1}^2 & \dots & 0 \\ \vdots & \ddots & \vdots \\ 0 & \dots & \sigma_{Y_k}^2 \end{bmatrix} - \begin{bmatrix} \mathrm{cov}(Y_1, Z_1+\epsilon) & \dots & 0 \\ \vdots & \ddots & \vdots \\ 0 & \dots & \mathrm{cov}(Y_k, Z_k+\epsilon) \end{bmatrix} \boldsymbol{\Sigma_{Z+\epsilon}^{-1}} \begin{bmatrix} \mathrm{cov}(Y_1, Z_1+\epsilon) & \dots & 0 \\ \vdots & \ddots & \vdots \\ 0 & \dots & \mathrm{cov}(Y_k, Z_k+\epsilon) \end{bmatrix} \right| \\
&= \left| \begin{bmatrix} \sigma_{Y_1}^2 - \mathrm{cov}^2(Y_1, Z_1+\epsilon)(\frac{1}{\sigma_{Z_1}^2} - \frac{\lambda}{\sigma_{Z_1}^4}) & \dots & \mathrm{cov}(Y_1, Z_1+\epsilon)\mathrm{cov}(Y_k, Z_k+\epsilon)\frac{\lambda}{\sigma_{Z_1}^2 \sigma_{Z_k}^2} \\ \vdots & & \vdots \\ \mathrm{cov}(Y_k, Z_k+\epsilon)\mathrm{cov}(Y_1, Z_1+\epsilon)\frac{\lambda}{\sigma_{Z_k}^2 \sigma_{Z_1}^2} & \dots & \sigma_{Y_k}^2 - \mathrm{cov}^2(Y_k, Z_k+\epsilon)(\frac{1}{\sigma_{Z_k}^2} - \frac{\lambda}{\sigma_{Z_k}^4}) \end{bmatrix} \right| \\
&= \left| \begin{bmatrix} \sigma_{Y_1}^2 - \frac{1}{\sigma_{Z_1}^2}\mathrm{cov}^2(Y_1, Z_1) & & \\ & \ddots & \\ & & \sigma_{Y_k}^2 - \frac{1}{\sigma_{Z_k}^2}\mathrm{cov}^2(Y_k, Z_k) \end{bmatrix} + \lambda \begin{bmatrix} \frac{1}{\sigma_{Z_1}^4}\mathrm{cov}^2(Y_1, Z_1) & \dots & \frac{1}{\sigma_{Z_1}^2 \sigma_{Z_k}^2}\mathrm{cov}(Y_1, Z_1)\mathrm{cov}(Y_k, Z_k) \\ \vdots & & \vdots \\ \frac{1}{\sigma_{Z_k}^2 \sigma_{Z_1}^2}\mathrm{cov}(Y_k, Z_k)\mathrm{cov}(Y_1, Z_1) & \dots & \frac{1}{\sigma_{Z_k}^4}\mathrm{cov}^2(Y_k, Z_k) \end{bmatrix} \right|
\end{aligned}
\tag{40}
$$

We use the notation $\boldsymbol{v} = \left[ \frac{1}{\sigma_{Z_1}^2}\mathrm{cov}(Y_1, Z_1) \quad \cdots \quad \frac{1}{\sigma_{Z_k}^2}\mathrm{cov}(Y_k, Z_k) \right]^T$, and $\boldsymbol{V} = \mathrm{diag}(\frac{1}{\sigma_{Z_1}^2}\mathrm{cov}^2(Y_1, Z_1), \cdots, \frac{1}{\sigma_{Z_k}^2}\mathrm{cov}^2(Y_k, Z_k))$. And utilize the rule of determinants of sums Marcus (1990), then we have:

$$
\begin{aligned}
|\boldsymbol{\Sigma_Y} - \boldsymbol{\Sigma_{Y(Z+\epsilon)}} \boldsymbol{\Sigma_{Z+\epsilon}^{-1}} \boldsymbol{\Sigma_{(Z+\epsilon)Y}}| &= |(\boldsymbol{\Sigma_Y} - \boldsymbol{V}) + \lambda \boldsymbol{v}\boldsymbol{v}^T| \\
&= |\boldsymbol{\Sigma_Y V}| + \lambda \boldsymbol{v}^T (\boldsymbol{\Sigma_Y} - \boldsymbol{V})^* \boldsymbol{v}
\end{aligned}
\tag{41}
$$

where $(\boldsymbol{\Sigma_Y} - \boldsymbol{V})^*$ is the adjoint of the matrix $(\boldsymbol{\Sigma_Y} - \boldsymbol{V})$. For simplicity, we can rewrite $|\boldsymbol{\Sigma_Y} - \boldsymbol{\Sigma_{Y(Z+\epsilon)}} \boldsymbol{\Sigma_{Z+\epsilon}^{-1}} \boldsymbol{\Sigma_{(Z+\epsilon)Y}}|$ as:

$$
\begin{aligned}
&|\boldsymbol{\Sigma_Y} - \boldsymbol{\Sigma_{Y(Z+\epsilon)}} \boldsymbol{\Sigma_{Z+\epsilon}^{-1}} \boldsymbol{\Sigma_{(Z+\epsilon)Y}}| \\
&= \prod_{i=1}^k (\sigma_{Y_i}^2 - \mathrm{cov}^2(Y_i, Z_i)\frac{1}{\sigma_{Z_i}^2}) + \Omega
\end{aligned}
\tag{42}
$$

where $\Omega = \lambda \boldsymbol{v}^T (\boldsymbol{\Sigma_Y} - \boldsymbol{V})^* \boldsymbol{v}$. The specific value of $\Omega$ can be obtained as:

$$
\Omega = \lambda \left[ \frac{1}{\sigma_{Z_1}^2}\mathrm{cov}(Y_1, Z_1) \quad \cdots \quad \frac{1}{\sigma_{Z_k}^2}\mathrm{cov}(Y_k, Z_k) \right] \begin{bmatrix} V_{11} & & \\ & \ddots & \\ & & V_{kk} \end{bmatrix} \begin{bmatrix} \frac{1}{\sigma_{Z_1}^2}\mathrm{cov}(Y_1, Z_1) \\ \vdots \\ \frac{1}{\sigma_{Z_k}^2}\mathrm{cov}(Y_k, Z_k) \end{bmatrix}
\tag{43}
$$

where the elements $V_{ii}, i \in [1, k]$ are minors of the matrix and expressed as:

$$
V_{ii} = \prod_{j=1, j\neq i}^k \left[ \sigma_{Y_j}^2 - \frac{1}{\sigma_{Z_j}^2}\mathrm{cov}^2(Z_j, Y_j) \right]
\tag{44}
$$

After some necessary steps, Equation 43 is reduced to:

$$
\begin{aligned}
\Omega &= \lambda \sum_{i=1}^k \frac{\frac{1}{\sigma_{Z_i}^4}\mathrm{cov}^2(Y_i, Z_i) \prod_{j=1}^k (\sigma_{Y_j}^2 - \mathrm{cov}^2(Y_j, Z_j)\frac{1}{\sigma_{Z_j}^2})}{(\sigma_{Y_i}^2 - \mathrm{cov}^2(Y_i, Z_i)\frac{1}{\sigma_{Z_i}^2})} \\
&= \lambda \prod_{i=1}^k (\sigma_{Y_i}^2 - \mathrm{cov}^2(Y_i, Z_i)\frac{1}{\sigma_{Z_i}^2}) \cdot \sum_{i=1}^k \frac{\mathrm{cov}^2(Z_i, Y_i)}{\sigma_{Z_i}^2(\sigma_{Z_i}^2 \sigma_{Y_i}^2 - \mathrm{cov}^2(Z_i, Y_i))}
\end{aligned}
\tag{45}
$$

Substitute Equation 45 into Equation 42, we can get:

$$
\begin{aligned}
&|\boldsymbol{\Sigma_Y} - \boldsymbol{\Sigma_{Y(Z+\epsilon)}} \boldsymbol{\Sigma_{Z+\epsilon}^{-1}} \boldsymbol{\Sigma_{(Z+\epsilon)Y}}| \\
&= \prod_{i=1}^k (\sigma_{Y_i}^2 - \mathrm{cov}^2(Y_i, Z_i)\frac{1}{\sigma_{Z_i}^2}) \cdot (1 + \lambda \sum_{i=1}^k \frac{\mathrm{cov}^2(Z_i, Y_i)}{\sigma_{Z_i}^2(\sigma_{Z_i}^2 \sigma_{Y_i}^2 - \mathrm{cov}^2(Z_i, Y_i))})
\end{aligned}
\tag{46}
$$

Accordingly, $|\mathbf{\Sigma_Y} - \mathbf{\Sigma_{YZ}}\mathbf{\Sigma_Z^{-1}}\mathbf{\Sigma_{ZY}}|$ is:

$$|\mathbf{\Sigma_Y} - \mathbf{\Sigma_{YZ}}\mathbf{\Sigma_Z^{-1}}\mathbf{\Sigma_{ZY}}| = \prod_{i=1}^{k}(\sigma_{Y_i}^2 - \frac{1}{\sigma_{Z_i}^2}\mathrm{cov}^2(Z_i, Y_i)) \tag{47}$$

As a result, $\frac{|\mathbf{\Sigma_{Y|Z+\epsilon}}|}{|\mathbf{\Sigma_{Y|Z}}|}$ is expressed as:

$$\frac{|\mathbf{\Sigma_{Y|Z}}|}{|\mathbf{\Sigma_{Y|Z+\epsilon}}|} = \frac{\prod_{i=1}^{k}(\sigma_{Y_i}^2 - \frac{1}{\sigma_{Z_i}^2}\mathrm{cov}^2(Z_i, Y_i))}{\prod_{i=1}^{k}(\sigma_{Y_i}^2 - \mathrm{cov}^2(Y_i, Z_i)\frac{1}{\sigma_{Z_i}^2}) \cdot (1 + \lambda\sum_{i=1}^{k}\frac{\mathrm{cov}^2(Z_i, Y_i)}{\sigma_{Z_i}^2(\sigma_{Z_i}^2\sigma_{Y_i}^2 - \mathrm{cov}^2(Z_i, Y_i))})} \tag{48}$$

Combine Equations 48 and 38 together, the entropy change is expressed as:

$$\triangle S(\mathcal{T}, \boldsymbol{\epsilon}) = \frac{1}{2}\log\frac{1}{(1 + \sigma_\epsilon^2\sum_{i=1}^{k}\frac{1}{\sigma_{Z_i}^2})(1 + \lambda\sum_{i=1}^{k}\frac{\mathrm{cov}^2(Z_i, Y_i)}{\sigma_{Z_i}^2(\sigma_{Z_i}^2\sigma_{Y_i}^2 - \mathrm{cov}^2(Z_i, Y_i))})} \tag{49}$$

It is difficult to tell that Equation 49 is greater or smaller than 0 directly. But one thing for sure is that when there is no Gaussian noise, Equation 49 equals 0. However, we can use another way to compare the numerator and denominator in Equation 49. Instead, we use the symbol $M$ to compare the numerator and denominator using subtraction. Let:

$$M = 1 - (1 + \sigma_\epsilon^2\sum_{i=1}^{k}\frac{1}{\sigma_{Z_i}^2})(1 + \lambda\sum_{i=1}^{k}\frac{\mathrm{cov}^2(Z_i, Y_i)}{\sigma_{Z_i}^2(\sigma_{Z_i}^2\sigma_{Y_i}^2 - \mathrm{cov}^2(Z_i, Y_i))}) \tag{50}$$

Obviously, the variance $\sigma_\epsilon^2$ of the Gaussian noise control the result of $M$, while the mean $\mu_\epsilon$ has no influence. When $\sigma_\epsilon$ approaching 0, we have:

$$\lim_{\sigma_\epsilon^2 \to 0} M = 0 \tag{51}$$

To determine if Gaussian noise can be positive noise, we need to determine whether the entropy change is large or smaller than 0.

$$\begin{cases} \triangle S(\mathcal{T}, \boldsymbol{\epsilon}) > 0 & \text{if } M > 0 \\ \triangle S(\mathcal{T}, \boldsymbol{\epsilon}) \leq 0 & \text{if } M \leq 0 \end{cases} \tag{52}$$

From the above equations, the sign of the entropy change is determined by the statistical properties of the embeddings and labels. Since $\epsilon^2 \geq 0$, $\lambda \geq 0$ and $\sum_{i=1}^{k}\frac{1}{\sigma_{Z_i}^2} \geq 0$, we need to have a deep dive into the residual part, i.e.,

$$\sum_{i=1}^{k}\frac{\mathrm{cov}^2(Z_i, Y_i)}{\sigma_{Z_i}^2(\sigma_{Z_i}^2\sigma_{Y_i}^2 - \mathrm{cov}^2(Z_i, Y_i))} = \sum_{i=1}^{k}\frac{\mathrm{cov}^2(Z_i, Y_i)}{\sigma_{Z_i}^4\sigma_{Y_i}^2(1 - \rho_{Z_iY_i}^2)} \tag{53}$$

where $\rho_{Z_iY_i}$ is the correlation coefficient, and $\rho_{Z_iY_i}^2 \in [0, 1]$. Eq. 53 is greater than 0, As a result, the sign of the entropy change in the Gaussian noise case is negative. We can conclude that Gaussian noise added to the latent space is harmful to the task.

## B.2 ADD GAUSSIAN NOISE TO RAW IMAGES

Assuming that the pixels of the raw images follow a Gaussian distribution. The variation of task complexity by adding Gaussian noise to raw images can be formulated as:

$$\begin{aligned} \triangle S(\mathcal{T}, \boldsymbol{\epsilon}) &= H(\mathcal{T}; \boldsymbol{X}) - H(\mathcal{T}; \boldsymbol{X} + \boldsymbol{\epsilon}) \\ &= \frac{1}{2}\log|\mathbf{\Sigma_{Y|X}}| - \frac{1}{2}\log|\mathbf{\Sigma_{Y|X+\epsilon}}| \\ &= \frac{1}{2}\log\frac{|\mathbf{\Sigma_{Y|X}}|}{|\mathbf{\Sigma_{Y|X+\epsilon}}|} \\ &= \frac{1}{2}\log\frac{|\mathbf{\Sigma_Y} - \mathbf{\Sigma_{YX}}\mathbf{\Sigma_X^{-1}}\mathbf{\Sigma_{XY}}|}{|\mathbf{\Sigma_Y} - \mathbf{\Sigma_{Y(X+\epsilon)}}\mathbf{\Sigma_{X+\epsilon}^{-1}}\mathbf{\Sigma_{(X+\epsilon)Y}}|} \\ &= \frac{1}{2}\log\frac{|\mathbf{\Sigma_Y} - \mathbf{\Sigma_{YX}}\mathbf{\Sigma_X^{-1}}\mathbf{\Sigma_{XY}}|}{|\mathbf{\Sigma_Y} - \mathbf{\Sigma_{YX}}\mathbf{\Sigma_{X+\epsilon}^{-1}}\mathbf{\Sigma_{XY}}|} \end{aligned} \tag{54}$$

Borrow the equations from the case of Gaussian noise added to the latent space, we have:

$$\triangle S(\mathcal{T}, \epsilon) = \frac{1}{2} \log \frac{1}{1 + \lambda \sum_{i=1}^{k} \frac{\text{cov}^2(X_i, Y_i)}{\sigma_{X_i}^2 (\sigma_{X_i}^2 \sigma_{Y_i}^2 - \text{cov}^2(X_i, Y_i))}} \tag{55}$$

Clearly, the introduction of Gaussian noise to each pixel in the original images has a detrimental impact on the task. **Note** that some studies have empirically shown that adding Gaussian noise to partial pixels of input images may be beneficial to the learning task Li (2022); Zhang et al. (2023).

## C  IMPACT OF LINEAR TRANSFORM NOISE ON TASK ENTROPY

In our work, concerning the image level perspective, "linear transform noise" denotes an image that is perturbed by another image or a combination of other images. From the viewpoint of embeddings, "linear transform noise" refers to an embedding perturbed by another embedding or the combination of other embeddings.

### C.1  INJECT LINEAR TRANSFORM NOISE INTO EMBEDDINGS

The entropy change of injecting linear transform noise into embeddings can be formulated as:

$$\begin{aligned}
\triangle S(\mathcal{T}, Q\mathbf{Z}) &= H(\mathcal{T}; \mathbf{Z}) - H(\mathcal{T}; \mathbf{Z} + Q\mathbf{Z}) \\
&= H(\mathbf{Y}; \mathbf{Z}) - H(\mathbf{Z}) - (H(\mathbf{Y}; \mathbf{Z} + Q\mathbf{Z}) - H(\mathbf{Z})) \\
&= H(\mathbf{Y}; \mathbf{Z}) - H(\mathbf{Y}; \mathbf{Z} + Q\mathbf{Z}) \\
&= \frac{1}{2} \log \frac{|\mathbf{\Sigma_Z}||\mathbf{\Sigma_Y} - \mathbf{\Sigma_{YZ}} \mathbf{\Sigma_Z^{-1}} \mathbf{\Sigma_{ZY}}|}{|\mathbf{\Sigma_{(I+Q)Z}}||\mathbf{\Sigma_Y} - \mathbf{\Sigma_{YZ}} \mathbf{\Sigma_Z^{-1}} \mathbf{\Sigma_{ZY}}|} \\
&= \frac{1}{2} \log \frac{1}{|I + Q|^2} \\
&= -\log |I + Q|
\end{aligned} \tag{56}$$

Since we want the entropy change to be greater than 0, we can formulate Equation 56 as an optimization problem:

$$\begin{aligned}
\max_{Q} &\triangle S(\mathcal{T}, Q\mathbf{Z}) \\
s.t. \ &rank(I + Q) = k \\
&[I + Q]_{ii} \geq [I + Q]_{ij}, i \neq j \\
&\|[I + Q]_i\|_1 = 1
\end{aligned} \tag{57}$$

The key to determining whether the linear transform is positive noise or not lies in the matrix of $Q$. The most important step is to ensure that $I + Q$ is invertible, which is $|(I + Q)| \neq 0$. For this, we need to investigate what leads $I + Q$ to be rank-deficient. The second constraint is to make the trained classifier get enough information about a specific embedding of an image and correctly predict the corresponding label. For instance, when an embedding $Z_1$ is perturbed by another embedding $Z_2$, the classifier predominantly relies on the information from $Z_1$ to predict the label $Y_1$. Conversely, if the perturbed embedding $Z_2$ takes precedence, the classifier struggles to accurately predict the label $Y_1$ and is more likely to predict it as label $Y_2$. The third constraint is the normalization of latent representations.

**Rank Deficiency Cases** To avoid causing a rank deficiency of $I + Q$, we need to figure out the conditions that lead to rank deficiency. Here we show a simple case causing the rank deficiency. When the matrix $Q$ is a backward identity matrix Horn & R. (2012),

$$Q_{i,j} = \begin{cases} 1, & i + j = k + 1 \\ 0, & i + j \neq k + 1 \end{cases} \tag{58}$$

i.e.,

$$Q = \begin{bmatrix} 0 & 0 & \ldots & 0 & 0 & 1 \\ 0 & 0 & \ldots & 0 & 1 & 0 \\ \vdots & \vdots & & \vdots & \vdots & \vdots \\ 0 & 1 & \ldots & 0 & 0 & 0 \\ 1 & 0 & \ldots & 0 & 0 & 0 \end{bmatrix} \tag{59}$$

then $(I + Q)$ will be:

$$I + Q = \begin{bmatrix} 1 & 0 & ... & 0 & 0 & 1 \\ 0 & 1 & ... & 0 & 1 & 0 \\ \vdots & \vdots & & \vdots & \vdots & \vdots \\ 0 & 1 & ... & 0 & 1 & 0 \\ 1 & 0 & ... & 0 & 0 & 1 \end{bmatrix} \tag{60}$$

Thus, $I + Q$ will be rank-deficient when $Q$ is a backward identity. In fact, when the following constraints are satisfied, the $I + Q$ will be rank-deficient:

$$\mathrm{HermiteForm}(I + Q)_i = \mathbf{0}, \quad \exists i \in [1, k] \tag{61}$$

where index $i$ is the row index, in this paper, the row index starts from 1, and $\mathrm{HermiteForm}$ is the Hermite normal form Kannan & Bachem (1979).

**Full Rank Cases** Except for the rank deficiency cases, $I + Q$ has full rank and is invertible. Since $Q$ is a row equivalent to the identity matrix, we need to introduce the three types of elementary row operations as follows Shores (2007).

> ▷ 1 **Row Swap** Exchange rows.
> Row swap here allows exchanging any number of rows. This is slightly different from the original one that only allows any two-row exchange since following the original row swap will lead to a rank deficiency. When the $Q$ is derived from $I$ with **Row Swap**, it will break the third constraint. Therefore, **Row Swap** merely is considered harmful and would degrade the performance of deep models.

> ▷ 2 **Scalar Multiplication** Multiply any row by a constant $\beta$. This breaks the fourth constraint, thus degrading the performance of deep models.

> ▷ 3 **Row Sum** Add a multiple of one row to another row. Then the matrix $I + Q$ would be like:

$$\begin{aligned} I + Q &= \begin{bmatrix} 1 & & & \\ & . & & \\ & & . & \\ & & & . & \\ & & & & 1 \end{bmatrix} + \begin{bmatrix} 1 & & & \\ & . & & \beta \\ & & . & \\ & & & . \\ & & & & 1 \end{bmatrix} \\ &= \begin{bmatrix} 2 & & & \\ & . & & \beta \\ & & . & \\ & & & . \\ & & & & 2 \end{bmatrix} \end{aligned} \tag{62}$$

> where $\beta$ can be at a random position beside the diagonal. As we can see from the simple example, **Row Sum** breaks the fourth constraint and makes entropy change smaller than 0.

From the above discussion, none of the single elementary row operations can guarantee positive effects on deep models.

However, if we combine the elementary row operations, it is possible to make the entropy change greater than 0 as well as satisfy the constraints. For example, we combine the **Row Sum** and **Scalar Multiplication** to generate the $Q$:

$$\begin{aligned} I + Q &= \begin{bmatrix} 1 & & & \\ & . & & \\ & & . & \\ & & & . \\ & & & & 1 \end{bmatrix} + \begin{bmatrix} -0.5 & 0.5 & & \\ & . & . & \\ & & . & . \\ & & & . & 0.5 \\ 0.5 & & & & -0.5 \end{bmatrix} \\ &= \begin{bmatrix} 0.5 & 0.5 & & \\ & . & . & \\ & & . & . \\ & & & . & 0.5 \\ 0.5 & & & & 0.5 \end{bmatrix} \end{aligned} \tag{63}$$

In this case, $\triangle S(\mathcal{T}, Q\boldsymbol{Z}) > 0$ when $Q = -0.5I$. The constraints are satisfied. This is just a simple case of adding linear transform noise that benefits deep models. Actually, there exists a design space of $Q$ that within the design space, deep models can reduce task entropy by injecting linear transform noise into the embeddings. To this end, we demonstrate that linear transform can be positive noise.

From the discussion in this section, we can draw conclusions that **Linear Transform Noise** can be positive under certain conditions, while **Gaussian Noise** and **Salt-and-pepper Noise** are harmful noise. From the above analysis, the conditions that satisfy positive noise form a design space. Exploring the design space of positive noise is an important topic for future work.

### C.1.1 OPTIMAL QUALITY MATRIX OF LINEAR TRANSFORM NOISE

The optimal quality matrix should maximize the entropy change and therefore theoretically define the minimized task complexity. The optimization problem as formulated in Equation 16 is:

$$
\begin{aligned}
\max_Q &- \log |I + Q| \\
s.t.\ &rank(I + Q) = k \\
&Q \sim I \\
&[I + Q]_{ii} \geq [I + Q]_{ij},\ i \neq j \\
&\|[I + Q]_i\|_1 = 1
\end{aligned}
\tag{64}
$$

Maximizing the entropy change is to minimize the determinant of the matrix sum of $I$ and $Q$. A simple but straight way is to design the matrix $Q$ that makes the elements in $I + Q$ equal, i.e.,

$$
I + Q = \begin{bmatrix} 1/k & \cdots & 1/k \\ \vdots & \cdots & \vdots \\ 1/k & \cdots & 1/k \end{bmatrix}
\tag{65}
$$

The determinant of the above equation is 0, but it breaks the first constraint of $rank(I + Q) = k$. However, by adding a small constant into the diagonal, and minus another constant by other elements, we can get:

$$
I + Q = \begin{bmatrix} 1/k + c_1 & \cdots & & 1/k - c_2 \\ 1/k - c_2 & \ddots & & \vdots \\ \vdots & & \ddots & 1/k - c_2 \\ 1/k - c_2 & \cdots & 1/k - c_2 & 1/k + c_1 \end{bmatrix}
\tag{66}
$$

Under the constraints, we can obtain the two constants that fulfill the requirements:

$$
c_1 = \frac{k - 1}{k(k + 1)}, \quad c_2 = \frac{1}{k(k + 1)}
\tag{67}
$$

Therefore, the corresponding $Q$ is:

$$
Q_{optimal} = \text{diag}\left(\frac{1}{k + 1} - 1, \ldots, \frac{1}{k + 1} - 1\right) + \frac{1}{k + 1}\mathbf{1}_{k \times k}
\tag{68}
$$

and the corresponding $I + Q$ is:

$$
I + Q = \begin{bmatrix} 2/(k + 1) & \cdots & & 1/(k + 1) \\ 1/(k + 1) & \ddots & & \vdots \\ \vdots & & \ddots & 1/(k + 1) \\ 1/(k + 1) & \cdots & 1/(k + 1) & 2/(k + 1) \end{bmatrix}
\tag{69}
$$

As a result, the determinant of optimal $I + Q$ can be obtained by following the identical procedure as Equation 41:

$$
|I + Q| = \frac{1}{(k + 1)^{k - 1}}
\tag{70}
$$

The upper boundary of entropy change of linear transform noise is determined:

$$
\triangle S(\mathcal{T}, Q\boldsymbol{Z})_{upper} = (k - 1)\log(k + 1)
\tag{71}
$$

## C.2 ADD LINEAR TRANSFORM NOISE TO RAW IMAGES

In this case, the task entropy with linear transform noise can be formulated as:

$$
\begin{aligned}
H(\mathcal{T}; \boldsymbol{X} + Q\boldsymbol{X}) &= -\sum_{\boldsymbol{Y} \in \mathcal{Y}} p(\boldsymbol{Y}|\boldsymbol{X} + Q\boldsymbol{X}) \log p(\boldsymbol{Y}|\boldsymbol{X} + Q\boldsymbol{X}) \\
&= -\sum_{\boldsymbol{Y} \in \mathcal{Y}} p(\boldsymbol{Y}|(I + Q)\boldsymbol{X}) \log p(\boldsymbol{Y}|(I + Q)\boldsymbol{X})
\end{aligned}
\tag{72}
$$

where $I$ is an identity matrix, and $Q$ is derived from $I$ using elementary row operations. Assuming that the pixels of the raw images follow a Gaussian distribution. The conditional distribution of $\boldsymbol{Y}$ given $\boldsymbol{X} + Q\boldsymbol{X}$ is also multivariate subjected to the normal distribution, which can be formulated as:

$$
\boldsymbol{Y}|(I + Q)\boldsymbol{X} \sim \mathcal{N}(\mathbb{E}(\boldsymbol{Y}|(I + Q)\boldsymbol{X}), var(\boldsymbol{Y}|(I + Q)\boldsymbol{X}))
\tag{73}
$$

Since the linear transform matrix is invertible, applying the linear transform to $\boldsymbol{X}$ does not alter the distribution of the $\boldsymbol{X}$. It is straightforward to obtain:

$$
\boldsymbol{\mu}_{\boldsymbol{Y}|(I+Q)\boldsymbol{X}} = \boldsymbol{\mu}_{\boldsymbol{Y}} + \boldsymbol{\Sigma}_{\boldsymbol{Y}\boldsymbol{X}} \boldsymbol{\Sigma}_{\boldsymbol{X}}^{-1} (I + Q)^{-1}((I + Q)X - (I + Q)\boldsymbol{\mu}_{\boldsymbol{X}})
\tag{74}
$$

$$
\boldsymbol{\Sigma}_{(\boldsymbol{Y}|(I+Q)\boldsymbol{X})} = \boldsymbol{\Sigma}_{\boldsymbol{Y}} - \boldsymbol{\Sigma}_{\boldsymbol{Y}\boldsymbol{X}} \boldsymbol{\Sigma}_{\boldsymbol{X}}^{-1} \boldsymbol{\Sigma}_{\boldsymbol{X}\boldsymbol{Y}}
\tag{75}
$$

Thus, the variation of task entropy adding linear transform noise can be formulated as:

$$
\begin{aligned}
\triangle S(\mathcal{T}, Q\boldsymbol{X}) &= H(\mathcal{T}; \boldsymbol{X}) - H(\mathcal{T}; \boldsymbol{X} + Q\boldsymbol{X}) \\
&= \frac{1}{2} \log |\boldsymbol{\Sigma}_{\boldsymbol{Y}|\boldsymbol{X}}| - \frac{1}{2} \log |\boldsymbol{\Sigma}_{\boldsymbol{Y}|\boldsymbol{X}+Q\boldsymbol{X}}| \\
&= \frac{1}{2} \log \frac{|\boldsymbol{\Sigma}_{\boldsymbol{Y}|\boldsymbol{X}}|}{|\boldsymbol{\Sigma}_{\boldsymbol{Y}|\boldsymbol{X}+Q\boldsymbol{X}}|} \\
&= \frac{1}{2} \log \frac{|\boldsymbol{\Sigma}_{\boldsymbol{Y}} - \boldsymbol{\Sigma}_{\boldsymbol{Y}\boldsymbol{X}} \boldsymbol{\Sigma}_{\boldsymbol{X}}^{-1} \boldsymbol{\Sigma}_{\boldsymbol{X}\boldsymbol{Y}}|}{|\boldsymbol{\Sigma}_{\boldsymbol{Y}} - \boldsymbol{\Sigma}_{\boldsymbol{Y}\boldsymbol{X}} \boldsymbol{\Sigma}_{\boldsymbol{X}}^{-1} \boldsymbol{\Sigma}_{\boldsymbol{X}\boldsymbol{Y}}|} \\
&= 0
\end{aligned}
\tag{76}
$$

The entropy change of 0 indicates that the implementation of linear transformation to the raw images could not help reduce the complexity of the task.

## D INFLUENCE OF SALT-AND-PEPPER NOISE ON TASK ENTROPY

Salt-and-pepper noise is a common type of noise that can occur in images due to various factors, such as signal transmission errors, faulty sensors, or other environmental factors Chan et al. (2005). Salt-and-pepper noise is often considered to be an independent process because it is a type of random noise that affects individual pixels in an image independently of each other Gonzales & Wintz (1987).

### D.1 INJECT SALT-AND-PEPPER NOISE INTO EMBEDDINGS

The entropy change of injecting salt-and-pepper noise can be formulated as:

$$
\begin{aligned}
\triangle S(\mathcal{T}, Q\boldsymbol{Z}) &= H(\mathcal{T}; \boldsymbol{Z}) - H(\mathcal{T}; \boldsymbol{Z}\boldsymbol{\epsilon}) \\
&= H(\boldsymbol{Y}; \boldsymbol{Z}) - H(\boldsymbol{Z}) - (H(\boldsymbol{Y}; \boldsymbol{Z}\boldsymbol{\epsilon}) - H(\boldsymbol{Z})) \\
&= H(\boldsymbol{Y}; \boldsymbol{Z}) - H(\boldsymbol{Y}; \boldsymbol{Z}\boldsymbol{\epsilon}) \\
&= -\sum_{\boldsymbol{Z} \in \mathcal{Z}} \sum_{\boldsymbol{Y} \in \mathcal{Y}} p(\boldsymbol{Z}, \boldsymbol{Y}) \log p(\boldsymbol{Z}, \boldsymbol{Y}) + \sum_{\boldsymbol{Z} \in \mathcal{Z}} \sum_{\boldsymbol{Y} \in \mathcal{Y}} \sum_{\boldsymbol{\epsilon} \in \mathcal{E}} p(\boldsymbol{Z}\boldsymbol{\epsilon}, \boldsymbol{Y}) \log p(\boldsymbol{Z}\boldsymbol{\epsilon}, \boldsymbol{Y}) \\
&= \mathbb{E} \left[ \log \frac{1}{p(\boldsymbol{Z}, \boldsymbol{Y})} \right] - \mathbb{E} \left[ \log \frac{1}{p(\boldsymbol{Z}\boldsymbol{\epsilon}, \boldsymbol{Y})} \right] \\
&= \mathbb{E} \left[ \log \frac{1}{p(\boldsymbol{Z}, \boldsymbol{Y})} \right] - \mathbb{E} \left[ \log \frac{1}{p(\boldsymbol{Z}, \boldsymbol{Y})} \right] - \mathbb{E} \left[ \log \frac{1}{p(\boldsymbol{\epsilon})} \right] \\
&= -\mathbb{E} \left[ \log \frac{1}{p(\boldsymbol{\epsilon})} \right] \\
&= -H(\boldsymbol{\epsilon})
\end{aligned}
\tag{77}
$$

Table 7: Details of ResNet Models. The columns "18-layer", "34-layer", "50-layer", and "101-layer" show the specifications of ResNet-18, ResNet-34, ResNet-50, and ResNet-101, separately.

| Layer name | Output size | 18-layer | 34-layer | 50-layer | 101-layer |
|---|---|---|---|---|---|
| conv1 | $112 \times 112$ | \multicolumn{4}{c}{$7 \times 7$, 64, stride 2} | | | |
| | | \multicolumn{4}{c}{$3 \times 3$, max pool, stride 2} | | | |
| conv2_x | $56 \times 56$ | $\begin{bmatrix} 3 \times 3 & 64 \\ 3 \times 3 & 64 \end{bmatrix} \times 2$ | $\begin{bmatrix} 3 \times 3 & 64 \\ 3 \times 3 & 64 \end{bmatrix} \times 3$ | $\begin{bmatrix} 1 \times 1 & 64 \\ 3 \times 3 & 64 \\ 1 \times 1 & 256 \end{bmatrix} \times 3$ | $\begin{bmatrix} 1 \times 1 & 64 \\ 3 \times 3 & 64 \\ 1 \times 1 & 256 \end{bmatrix} \times 3$ |
| conv3_x | $28 \times 28$ | $\begin{bmatrix} 3 \times 3 & 128 \\ 3 \times 3 & 128 \end{bmatrix} \times 2$ | $\begin{bmatrix} 3 \times 3 & 128 \\ 3 \times 3 & 128 \end{bmatrix} \times 4$ | $\begin{bmatrix} 1 \times 1 & 128 \\ 3 \times 3 & 128 \\ 1 \times 1 & 512 \end{bmatrix} \times 4$ | $\begin{bmatrix} 1 \times 1 & 128 \\ 3 \times 3 & 128 \\ 1 \times 1 & 512 \end{bmatrix} \times 4$ |
| conv4_x | $14 \times 14$ | $\begin{bmatrix} 3 \times 3 & 256 \\ 3 \times 3 & 256 \end{bmatrix} \times 2$ | $\begin{bmatrix} 3 \times 3 & 256 \\ 3 \times 3 & 256 \end{bmatrix} \times 6$ | $\begin{bmatrix} 1 \times 1 & 256 \\ 3 \times 3 & 256 \\ 1 \times 1 & 1024 \end{bmatrix} \times 6$ | $\begin{bmatrix} 1 \times 1 & 256 \\ 3 \times 3 & 256 \\ 1 \times 1 & 1024 \end{bmatrix} \times 23$ |
| conv5_x | $7 \times 7$ | $\begin{bmatrix} 3 \times 3 & 512 \\ 3 \times 3 & 512 \end{bmatrix} \times 2$ | $\begin{bmatrix} 3 \times 3 & 512 \\ 3 \times 3 & 512 \end{bmatrix} \times 3$ | $\begin{bmatrix} 1 \times 1 & 512 \\ 3 \times 3 & 512 \\ 1 \times 1 & 2048 \end{bmatrix} \times 3$ | $\begin{bmatrix} 1 \times 1 & 512 \\ 3 \times 3 & 512 \\ 1 \times 1 & 2048 \end{bmatrix} \times 3$ |
| | $1 \times 1$ | \multicolumn{4}{c}{average pool, 1000-d fc, softmax} | | | |
| Params | | 11M | 22M | 26M | 45M |

The entropy change is smaller than 0, therefore, the salt-and-pepper is a pure detrimental noise to the learning task.

### D.2 ADD SALT-AND-PEPPER NOISE TO RAW IMAGES

The task entropy with salt-and-pepper noise is rewritten as:

$$H(\mathcal{T}; \boldsymbol{X}\boldsymbol{\epsilon}) = - \sum_{\boldsymbol{Y} \in \mathcal{Y}} p(\boldsymbol{Y}|\boldsymbol{X}\boldsymbol{\epsilon}) \log p(\boldsymbol{Y}|\boldsymbol{X}\boldsymbol{\epsilon}) \tag{78}$$

Since $\boldsymbol{\epsilon}$ is independent of $\boldsymbol{X}$ and $\boldsymbol{Y}$, the above equation can be expanded as:

$$\begin{aligned} H(\mathcal{T}; \boldsymbol{X}\boldsymbol{\epsilon}) &= - \sum_{\boldsymbol{Y} \in \mathcal{Y}} \frac{p(\boldsymbol{Y}, \boldsymbol{X}\boldsymbol{\epsilon})}{p(\boldsymbol{X})p(\boldsymbol{\epsilon})} \log \frac{p(\boldsymbol{Y}, \boldsymbol{X}\boldsymbol{\epsilon})}{p(\boldsymbol{X})p(\boldsymbol{\epsilon})} \\ &= - \sum_{\boldsymbol{Y} \in \mathcal{Y}} \frac{p(\boldsymbol{Y}, \boldsymbol{X})p(\boldsymbol{\epsilon})}{p(\boldsymbol{X})p(\boldsymbol{\epsilon})} \log \frac{p(\boldsymbol{Y}, \boldsymbol{X})p(\boldsymbol{\epsilon})}{p(\boldsymbol{X})p(\boldsymbol{\epsilon})} \\ &= - \sum_{\boldsymbol{Y} \in \mathcal{Y}} p(\boldsymbol{Y}|\boldsymbol{X}) \log p(\boldsymbol{Y}|\boldsymbol{X}) \end{aligned} \tag{79}$$

where

$$\begin{aligned} p(\boldsymbol{X}\boldsymbol{\epsilon}, \boldsymbol{Y}) &= p(\boldsymbol{X}\boldsymbol{\epsilon}|\boldsymbol{Y})p(\boldsymbol{Y}) \\ &= p(\boldsymbol{X}|\boldsymbol{Y})p(\boldsymbol{\epsilon}|\boldsymbol{Y})p(\boldsymbol{Y}) \\ &= p(\boldsymbol{X}|\boldsymbol{Y})p(\boldsymbol{\epsilon})p(\boldsymbol{Y}) \\ &= p(\boldsymbol{X}, \boldsymbol{Y})p(\boldsymbol{\epsilon}) \end{aligned} \tag{80}$$

Therefore, the entropy change with salt-and-pepper noise is:

$$\triangle S(\mathcal{T}, Q\boldsymbol{X}) = H(\mathcal{T}; \boldsymbol{X}) - H(\mathcal{T}; \boldsymbol{X}\boldsymbol{\epsilon}) = 0 \tag{81}$$

Salt-and-pepper noise can not help reduce the complexity of the task, and therefore, it is considered a type of pure detrimental noise.

## E EXPERIMENTAL SETTING

In this section, we present the implementation details. The noise was added during both the training and inference stages. Model details of the models are shown in Table 7 and 8. Pre-trained models on ImageNet-21K are used. We train all ResNet and ViT-based models using AdamW optimizer Loshchilov & Hutter (2017). We set the learning rate of each parameter group using a

Table 8: Details of ViT Models. Each row shows the specifications of a kind of ViT model. ViT-T, ViT-S, ViT-B, and ViT-L represent ViT Tiny, ViT Small, ViT Base, and ViT Large, separately.

| ViT Model | Layers | Hidden size | MLP size | Heads | Params |
|-----------|--------|-------------|----------|-------|--------|
| ViT-T | 12 | 192 | 768 | 3 | 5.7M |
| ViT-S | 12 | 384 | 1536 | 6 | 22M |
| ViT-B | 12 | 768 | 3072 | 12 | 86M |
| ViT-L | 24 | 1024 | 4096 | 16 | 307M |

Table 9: Top 1 accuracy on ImageNet V2 with positive linear transform noise.

| Model | Top1 Acc. | Params. | Image Res. | Pretrained Dataset |
|-------|-----------|---------|------------|--------------------|
| ViT-B | 72.6 | 86M | $224 \times 224$ | ImageNet 21k |
| NoisyNN (ViT-B based) | 82.2 | 86M | $224 \times 224$ | ImageNet 21k |
| NoisyNN (ViT-B based) | 84.8 | 86M | $384 \times 384$ | ImageNet 21k |

cosine annealing schedule with a minimum of $1e - 7$. Data are resized and then normalized before passing into the model.

**CNN (ResNet) Setting** The training epoch is set to 100. We initialized the learning rate as 0 and linearly increase it to 0.001 after 10 warmup steps. All the experiments of CNNs are trained on a single Tesla V100 GPU with 32 GB. The batch size for ResNet18, ResNet34, ResNet50, and ResNet101 are 1024, 512, 256, and 128, respectively.

**ViT and Variants Setting** All the experiments of ViT and its variants are trained on a single machine with 8 Tesla V100 GPUs. For vanilla ViTs, including ViT-T, ViT-S, ViT-B, and ViT-L, the training epoch is set to 50 and the input patch size is $16 \times 16$. We initialized the learning rate as 0 and linearly increase it to 0.0001 after 10 warmup steps. We then decrease it by the cosine decay strategy. For experiments on the variants of ViT, the training epoch is set to 100 and the learning rate is set to 0.0005 with 10 warmup steps.

## F    MORE EXPERIMENT RESULTS

### F.1    IMAGENETV2 RESULTS

Table 9 shows additional results on ImageNetV2. We tested the positive linear transformation noise on ImageNetV2, and these results demonstrate the superiority of our proposed methods.

### F.2    IMAGENET-A RESULTS

Table 10 shows additional results on ImageNet-A. We further tested the positive linear transformation noise on ImageNet-A, which exhibits a significant domain shift compared to the validation set of ImageNet-1k. The results demonstrate the robustness of our method to domain shift. We also calculate the confusion matrices of our method and ViT-B on ImageNet-A, which are presented in Fig. 3 and 4, respectively.

Table 10: Top 1 accuracy on ImageNet-A with positive linear transform noise.

| Model | Top1 Acc. | Params. | Image Res. | Pretrained Dataset |
|-------|-----------|---------|------------|--------------------|
| ViT-B | 27.4 | 86M | $224 \times 224$ | ImageNet 21k |
| NoisyNN (ViT-B based) | 34.1 | 86M | $224 \times 224$ | ImageNet 21k |
| NoisyNN (ViT-B based) | 38.3 | 86M | $384 \times 384$ | ImageNet 21k |

### F.3    IMAGENET-C RESULTS

Table 11 shows additional results on ImageNet-C. ImageNet-C exhibits various forms of domain shift in comparison to the validation set of ImageNet-1k. The results further demonstrate the robustness of our method to such domain shifts.

Table 11:  Top 1 accuracy on ImageNet-C with positive linear transform noise.

| Model | Top1 Acc. | Params. | Image Res. | Pretrained Dataset |
|---|---|---|---|---|
| ViT-B | 53.4 | 86M | $224 \times 224$ | ImageNet 21k |
| NoisyNN (ViT-B based) | 58.1 | 86M | $224 \times 224$ | ImageNet 21k |
| NoisyNN (ViT-B based) | 60.5 | 86M | $384 \times 384$ | ImageNet 21k |

### F.4    TINYIMAGENET RESULTS

Results on TinyImageNet are shown in Table 12 and 13. These results further confirm our analysis in the main paper that Gaussian Noise and Salt-and-pepper Noise are harmful noise, while Linear Transform Noise can be made positive noise. Note that even with extensive hyperparameter search, Gaussian noise (Table 14) and salt-and-pepper noise (Table 15) still substaintially under-perform positive linear transform noise.

### F.5    CIFAR AND INBREAST RESULTS

Results on CIFAR-10, CIFAR-100, and INbreast are shown in Table 16. Showing the effectiveness of NoisyNN beyond ImageNet-based datasets.

### F.6    COMPARISON AND COMBINATION WITH COMMON DATA AUGMENTATION TECHNIQUES

We compare our method with common data augmentation methods, and the results are presented in Table 18. Additionally, we combine our method with data augmentations, and the corresponding results are shown in Table 17.

### F.7    COMPARISON WITH OTHER NOISES

Below in Table 19 we compare NoisyNN to other commonly seen noises including White Noise, Uniform Noise and Dropout (Srivastava et al., 2014) on TinyImageNet.

### F.8    COMPARISON WITH MANIFOLD MIXUP

Beside the key differences discussed in the main paper, other difference between NoisyNN and Manifold MixUp include: Manifold MixUp introduces randomness in the strength of interpolation by drawing from a probability distribution, whereas we use a fixed strength based on theoretical guidance. Under the constraint of Eq 16, a larger $\alpha$ induces a more substantial entropy change in Eq 15, as verified by Figure 2 (a) (c). Additionally, Manifold MixUp selects random mixing layers during training, while we use a fixed layer (chosen before training and kept fixed). In our experiments, we use the last layer, with an ablation study on the effect of choosing different layers. Below in

Table 12: ResNet with different kinds of noise on TinyImageNet. Vanilla means the vanilla model without noise. Accuracy is shown in percentage. Gaussian noise used here is subjected to standard normal distribution. Linear transform noise used in this table is designed to be positive noise. The difference is shown in the bracket.

| Model | ResNet-18 | ResNet-34 | ResNet-50 | ResNet-101 |
|---|---|---|---|---|
| Vanilla | 64.01 (+0.00) | 67.04 (+0.00) | 69.47 (+0.00) | 70.66 (+0.00) |
| + Gaussian Noise | 63.23 (-0.78) | 65.71 (-1.33) | 68.17 (-1.30) | 69.13 (-1.53) |
| + Linear Transform Noise | **73.32 (+9.31)** | **76.70 (+9.66)** | **76.88 (+7.41)** | **77.30 (+6.64)** |
| + Salt-and-pepper Noise | 55.97 (-8.04) | 63.52 (-3.52) | 49.42 (-20.25) | 53.88 (-16.78) |

Table 13: ViT with different kinds of noise on TinyImageNet. Vanilla means the vanilla model without injecting noise. Accuracy is shown in percentage. Gaussian noise used here is subjected to standard normal distribution. Linear transform noise used in this table is designed to be positive noise. The difference is shown in the bracket. Note **ViT-L is overfitting on TinyImageNet** Dosovitskiy et al. (2020) Steiner et al. (2021).

| Model | ViT-T | ViT-S | ViT-B | ViT-L |
|---|---|---|---|---|
| Vanilla | 81.75 (+0.00) | 86.78 (+0.00) | 90.48 (+0.00) | 93.32 (+0.00) |
| + Gaussian Noise | 80.95 (-0.80) | 85.66 (-1.12) | 89.61 (-0.87) | 92.31 (-1.01) |
| + Linear Transform Noise | **82.50 (+0.75)** | **91.62 (+4.84)** | **94.92 (+4.44)** | **93.63 (+0.31)** |
| + Salt-and-pepper Noise | 79.34 (-2.41) | 84.66 (-2.12) | 87.45 (-3.03) | 83.48 (-9.84) |

Table 14: Impact of Different Combinations of Mean and Standard Deviation of Gaussian Noise on TinyImageNet Performance with ViT-S.

| Gaussian Noise (Mean, STD) | TinyImageNet |
|---|---|
| (0, 0.5) | 86.8 |
| (0, 1.0) | 85.9 |
| (1.0, 0.5) | 86.4 |
| (1.0, 1.0) | 85.7 |
| NoisyNN | 91.6 |

Table 20 we compare NoisyNN to Manifold MixUp (Verma et al., 2019) and verify the design choice of using fixed layer versus random layer during training. The results show that NoisyNN achieves better performance. Experiments conducted on on TinyImageNet.

### F.9 Domain Generalization

Domain Generalization (DG) methods try to learn a robust model by training on multiple source domains Volpi et al. (2018); Seo et al. (2020); Carlucci et al. (2019); Huang et al. (2020), while DG methods cannot access the target domains during the training stage. To verify our method in the application of DG tasks, we further conduct experiments on VLCS and PACS, two commonly used datasets in the field of DG. The results are reported in Table 21. As shown in the table, compared to competitive methods, our proposed method achieves state-of-the-art (SOTA) results on the PACS and VLCS datasets.

### F.10 Text Classification

Text classification involves categorizing text into predefined classes or labels (Kowsari et al., 2019). It is widely used in various applications such as spam detection, sentiment analysis, topic labeling, and document categorization. To check whether our method can be applied to a different data modality but within the same problem of classification, we conduct experiments on two popular text classification datasets with widely used models. The results are shown in Table 22. Equipped with our method, TextCNN and TextRNN show a significant improvement in performance.

Table 15: Impact of Salt-and-Pepper Noise on TinyImageNet Performance with ViT-S.

| Salt-and-Pepper Noise (Intensity) | TinyImageNet |
|---|---|
| 0.1 | 86.0 |
| 0.2 | 85.4 |
| 0.3 | 84.6 |
| 0.4 | 83.5 |
| NoisyNN | 91.6 |

Table 16: Comparing ViT-B with NoisyNN on CIFAR-10, CIFAR-100 and INbreast.

| Model | CIFAR-100 | CIFAR-10 | INbreast |
|---|---|---|---|
| ViT-B | 91.5±0.1 | 98.6±0.1 | 90.6±0.2 |
| NoisyNN (ViT-B based) | 93.7±0.1 | 99.4±0.1 | 93.5±0.1 |

Table 17: Combining NoisyNN with Data Augmentation.

| Method | ImageNet |
|---|---|
| NoisyNN (No DA) | 89.9±0.5 |
| NoisyNN + RandomResizedCrop | 89.1±0.5 |
| NoisyNN + RandomHorizontalFlip+RandomResizedCrop | 89.2±0.6 |
| NoisyNN + RandomResizedCrop+RandAugment | 89.4±0.5 |

Table 18: Comparing NoisyNN with Data Augmentation.

| Method | ImageNet |
|---|---|
| ViT-B | 84.3 |
| ViT-B+RandomFlip+Gaussian Blur | 84.2 |
| ViT-B+RandAugment | 85.1 |
| ViT-B+Linear Transformation Noise (NoisyNN) | **89.9** |

Table 19: Comparison of NoisyNN with other noises on TinyImageNet.

| | ResNet18 | ResNet34 | ResNet50 |
|---|---|---|---|
| Vanilla | 64.01 | 67.04 | 69.47 |
| White Noise | 64.05 | 65.97 | 68.87 |
| Uniform Noise | 64.05 | 66.01 | 69.01 |
| Gaussian Noise | 63.23 | 64.71 | 68.17 |
| Salt-and-pepper | 55.97 | 63.52 | 49.42 |
| Dropout | 63.96 | 67.01 | 69.40 |
| NoisyNN (ours) | **73.32** | **76.70** | **76.88** |

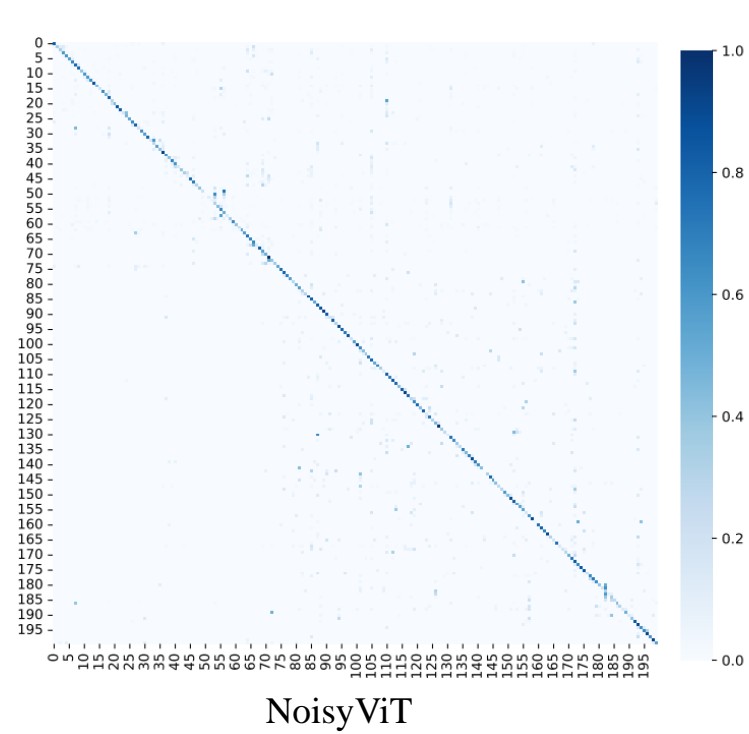



NoisyViT



Figure 3: Confusion Matrix of NoisyNN (ViT-based) on ImageNet-A.

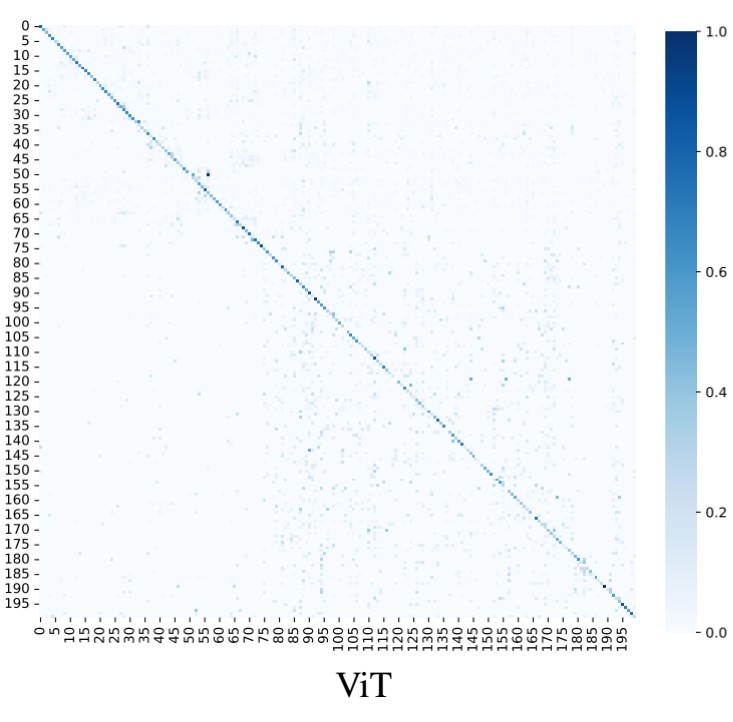



ViT



Figure 4: Confusion Matrix of ViT on ImageNet-A.

Table 20: Comparison with Manifold MixUp on TinyImageNet

|                        | ResNet18 | ResNet34 | ViT-S | ViT-B |
|------------------------|----------|----------|-------|-------|
| Vanilla                | 64.01    | 67.04    | 86.78 | 90.48 |
| Manifold Mixup         | 71.83    | 75.28    | 89.87 | 93.21 |
| NoisyNN (random layer) | 72.29    | 75.88    | 90.02 | 93.76 |
| NoisyNN (default)      | **73.32**| **76.70**| **91.62**| **94.92** |

Table 21: Comparison with other methods in domain generalization tasks.

| Method                                   | PACS | VLCS |
|------------------------------------------|------|------|
| ViT Dosovitskiy et al. (2020) (ICLR'21)  | 85.0 | 76.9 |
| SDViT (Sultana et al., 2022) (ACCV'22)   | 88.9 | 81.9 |
| ALOFT (Guo et al., 2023) (CVPR'23)       | 91.6 | 81.3 |
| NoisyViT                                 | 93.1 | 84.4 |

## F.11 OBJECT DETECTION

Here we explored the NoisyNN framework for object detection tasks. The preliminary experiments in Table 23 show the promise of extending the NoisyNN framework for Object Detection tasks. Experiments conducted on COCO dataset (Lin et al., 2014).

## F.12 COMPUTATIONAL OVERHEAD

Our NoisyNN does not incur additional computation costs beyond a simple linear transformation in the embedding space. Below in Table 24 we show the runtime comparison.

## G DOMAIN ADAPTATION DETAILS

Unsupervised domain adaptation (UDA) aims to learn transferable knowledge across the source and target domains with different distributions Pan & Yang (2009); Wei et al. (2018). There are mainly two kinds of deep neural networks for UDA, which are CNN-based and Transformer-based methods Sun et al. (2022); Yang et al. (2023a). Various techniques for UDA are adopted on these backbone architectures. For example, the discrepancy techniques measure the distribution divergence between source and target domains Long et al. (2018); Sun & Saenko (2016). Adversarial adaptation discriminates domain-invariant and domain-specific representations by playing an adversarial game between the feature extractor and a domain discriminator Ganin & Lempitsky (2015).

Recently, transformer-based methods achieved SOTA results on UDA, therefore, we evaluate the ViT-B with the positive noise on widely used UDA benchmarks. Here the positive noise is the linear transform noise identical to that used in the classification task. The positive noise is injected into the embeddings of the last layer of the model, mirroring the same setting taken in the classification task. The datasets include **Office Home** Venkateswara et al. (2017) and **VisDA2017** Peng et al. (2017). **Office-Home** Venkateswara et al. (2017) has 15,500 images of 65 classes from four domains: Artistic (Ar), Clip Art (Cl), Product (Pr), and Real-world (Rw) images. **VisDA2017** is a Synthetic-to-Real object recognition dataset, with more than 0.2 million images in 12 classes. We use the ViT-B with a $16 \times 16$ patch size, pre-trained on ImageNet. We use minibatch Stochastic Gradient Descent (SGD) optimizer Ruder (2016) with a momentum of 0.9 as the optimizer. The batch size is set to 32. We initialized the learning rate as 0 and linearly warm up to 0.05 after 500 training steps. The results

Table 22: Comparison with other methods in text classification tasks.

| Method                               | THUNews | AGNews |
|--------------------------------------|---------|--------|
| TextCNN (Kim, 2014) (EMNLP'14)       | 90.8    | 89.2   |
| NoisyTextCNN                         | 93.4    | 89.3   |
| TextRNN (Liu et al., 2016) (IJCAI'16)| 90.7    | 87.7   |
| NoisyTextRNN                         | 95.5    | 88.1   |

Table 23: Object Detection with the NoisyNN framework on COCO dataset.

|           | DETR | NoisyDETR |
|-----------|------|-----------|
| $AP$      | 42.0 | 42.7      |
| $AP_{50}$ | 62.4 | 62.9      |
| $AP_{75}$ | 44.2 | 44.8      |
| $AP_S$    | 20.5 | 21.4      |
| $AP_M$    | 45.8 | 45.9      |
| $AP_L$    | 61.1 | 62.0      |

Table 24: Runtime Comparison between NoisyViT and ViT on ImageNet.

| Machine                            | ViT         | NoisyViT    |
|------------------------------------|-------------|-------------|
| Nvidia TITAN, Ubuntu, Intel i7-9700K | 2h43m/epoch | 2h45m/epoch |

are shown in Table 4 and 5. The methods above the black line are based on CNN architecture, while those under the black line are developed from the Transformer architecture. The NoisyTVT-B, i.e., TVT-B with positive noise, achieves better performance than existing works. These results show that positive noise also works in domain adaptation tasks.

