# OpenReview forum: "Noise-Augmented Deep Neural Networks for Image Classification: Insights from Information Theory"
_ICLR.cc/2025/Conference — Submitted to ICLR 2025_

### Official Review · Reviewer_xiNE · 2024-10-28

**Soundness:** 1
**Presentation:** 2
**Contribution:** 2
**Rating:** 1
**Confidence:** 4

**Summary:**

The paper introduces NoisyNN, a novel approach that leverages positive noise to improve deep neural networks' performance on tasks like image classification and domain adaptation. The authors categorize noise into positive noise (PN) and harmful noise (HN) based on their effect on task complexity, using information-theoretic metrics. The analysis is conducted on various noise type including Linear transformation noise, Gaussian and salt-and-pepper noise. Extensive experiments validate NoisyNN's effectiveness using CNNs and Vision Transformers (ViTs) across multiple datasets.

However, a major red flag is the suspicious ImageNet-1K accuracy. No model (22K pretrained + 1K finetuned) has ever achieved 95% top-1 accuracy on the ImageNet-1K validation set. Even with extensive web-scale pre-training, the best state-of-the-art performance remains below 92.5% (https://paperswithcode.com/sota/image-classification-on-imagenet). I intend to reproduce the reported performance. However, the README in the anonymous submission is completely missing, raising concerns about the completeness of the codebase. This lack of professionalism further reflects the questionable nature of your reported performance. I am willing to proceed, but please provide detailed instructions first. Given the unusually high performance you reported, you should be thoroughly familiar with the codebase, so there should be no reason for not providing this.

**Strengths:**

- (1) The analysis of noise choices is thorough including Linear transformation noise, Gaussian and salt-and-pepper noise.

- (2) The method can be generalized to various architecture includes ResNet and ViT.

**Weaknesses:**

- (1) Poor writing organization. The authors study three types of noise but the linear transformation noise is the one with positive effect. Thus, the authors are suggest to emphasize that the proposed NoisyNN is built upon linear  linear Transformation noise to avoid confusion.

- (2) Experimental details are not explained clearly.

    - (a) The author report the results of Vanilla ViT-B with accuracy of 84.33%. It would be great if the author could reference the codebase/implementation of the baseline.

    - (b) Are the noises applied to the imageNet-1k fine-tuning or {imageNet21K pretraining + imageNet1K fine-tuning} ?

- (3) The specifics of injecting noise at the feature or feature-grid level are unclear. Is the noise applied to each feature grid individually?

- (4) The comparison with state-of-the-art data augmentation models is missing. NoisyNN is essentially a data augmentation technique, but the related work in Section 2 only covers research up to 2020—prior to the introduction of vision Transformers. As a result, the references are outdated. Additionally, the approaches compared in Table 18 are overly simplistic, relying on basic techniques such as “RandomFlip + Gaussian Blur + RandAugment.”

- (4) Suspicious Claim of Accuracy in ImageNet-1K. The reasons are explained in (a) (b) (c) below. The suggestion is provided in (d) below.


    - (a) One of the major red flags is the suspiciously high ImageNet-1K top-1 accuracy reported in this paper. In Table 3, the proposed approach claims to achieve 94.8% top-1 accuracy. However, no model has ever reached such a performance on the ImageNet-1K validation set. Even with extensive web-scale pre-training, the best state-of-the-art performance remains below 92.5% ([https://paperswithcode.com/sota/image-classification-on-imagenet](https://paperswithcode.com/sota/image-classification-on-imagenet)). This discrepancy is a critical concern for the computer vision community.

    - (b) Under the same imageNet21K pretraing and imageNet1K fine-tuning setting, the state-of-the-art computer vision model is CoAtNet-4 (Table 4 in ) achieving 88.56%.

    - (c) Even with stronger web-scale image-text pre-training, PaLi [2] and CoCa [3] can only reach 90.9% and 91.0% respectively on imageNet-1K.

    - (d) The authors are suggested to provide more constructive guideline on the reproduction of the codebase as raise in the Question section.

- (5) I am going to reproduce the performance. But the README in the anonymous is entirely MISSING, and thus I have concerns about the completeness of the codebase. I am willing to reproduce the code, but first, please provide clear instructions. There should be no excuse, as you are presumably very familiar with the codebase, given the unusually high performance you achieved.

[1] Dai Z, Liu H, Le Q V, et al. Coatnet: Marrying convolution and attention for all data sizes[J]. Advances in neural information processing systems, 2021, 34: 3965-3977 (1283 citations till Oct 2024).

[2] Chen X, Wang X, Changpinyo S, et al. Pali: A jointly-scaled multilingual language-image model[J]. ICLR, 2023 (567 citations till Oct 2024).

[3] Yu J, Wang Z, Vasudevan V, et al. Coca: Contrastive captioners are image-text foundation models[J]. TMLR, 2022 (1225 citations till Oct 2024).

[4] Tu Z, Talebi H, Zhang H, et al. Maxvit: Multi-axis vision transformer[C]//European conference on computer vision. Cham: Springer Nature Switzerland, 2022: 459-479 (620 citations till Oct 2024).

[5] Liu Z, Mao H, Wu C Y, et al. A convnet for the 2020s[C]//Proceedings of the IEEE/CVF conference on computer vision and pattern recognition. 2022: 11976-11986 (5765 citations till Oct 2024).

**Questions:**

- Can the authors assess the transferability of their model to downstream object detection and segmentation tasks. In practice, models trained on ImageNet [4, 5] often utilize ImageNet1K-pretrained weights as the pretrained backbone for these downstream applications.

- I plan to reproduce the performance, but the README (reproduction instructions) in the anonymous submission is completely missing. This raises concerns about the codebase's completeness. The author should include detailed instructions, covering installation, training, and entry scripts. Without these, reproducibility is impossible. Even if the codebase of this work may be built upon timm and opensource codebase, the authors should still provide the reproducing instruction. Here’s a step-by-step guide to help you.

    - Installation instruction, please refer to the template([https://github.com/facebookresearch/ConvNeXt/blob/main/INSTALL.md](https://github.com/facebookresearch/ConvNeXt/blob/main/INSTALL.md))

    - Training instruction, and command with all necessary arguments, please refer to the template ([https://github.com/facebookresearch/ConvNeXt/blob/main/INSTALL.md](https://github.com/facebookresearch/ConvNeXt/blob/main/INSTALL.md)).  More systematical version can be seen in timm.

    - Evaluation procedure

    - Data preparation steps. Was the validation data separated from the training data?

    - Release of the model weights

    - Details for the packages and the version.

**Details Of Ethics Concerns:**

A major red flag is the suspicious ImageNet-1K accuracy. No model has ever achieved 95% top-1 accuracy on the ImageNet-1K validation set. Even with extensive web-scale pre-training, the best state-of-the-art performance remains below 92.5% (https://paperswithcode.com/sota/image-classification-on-imagenet). I intend to reproduce the reported performance. However, the README in the anonymous submission is completely missing, raising concerns about the completeness of the codebase. This lack of professionalism further reflects the questionable nature of your reported performance.

---

### Official Review · Reviewer_4wNv · 2024-11-01

**Soundness:** 3
**Presentation:** 2
**Contribution:** 2
**Rating:** 5
**Confidence:** 2

**Summary:**

This paper analyzes the proactive noise injection into deep learning models from both theoretical and empirical perspectives. This paper theoretically characterizes the positive noise that reduces task complexity based on information entropy, and demonstrates the presence of positive noise through experiments. The approach ‘NoisyNN’ leveraging positive noise is proposed, and its efficacy is shown through empirical experiments.

**Strengths:**

1. Theoretical analysis is provided to characterize the idea of 'positive noise' proposed in this paper.

2. Empirical experiments are given in this paper to support the advantage of leveraging positive noise.

3. The approach 'NoisyNN' that utilizes the positive noise is proposed in this paper, and its efficacy is demonstrated through empirical experiments.

**Weaknesses:**

1. This paper introduced the‘NoisyNN’ method based on a specific type of positive noise, namely the positive linear transformation noise, and the following empirical experiments are also for the 'NoisyNN' method. How does the idea of this paper work with other types of positive noise?

2. This paper aims to analyze the influence of injecting noise both theoretically and empirically. The empirical experiments focus on image classification and domain adaptation, alongside several other related tasks. However, the current theoretical analysis is primarily for classification tasks. How might the current theoretical analysis be extended to cover other related tasks mentioned in this paper, for example, those mentioned in Section 4?

**Questions:**

1. Apart from the specific types of noise analyzed in this paper (Section 3), how do other types of positive noise work under the idea of this paper?

2. The novelty of this paper seems mainly lies in proposing the concept of 'positive noise' and its application. How does this paper compare with existing papers with similar ideas of adding noise? And the novelty of this paper compared to those papers?

---

### Official Review · Reviewer_vxMD · 2024-11-04

**Soundness:** 3
**Presentation:** 3
**Contribution:** 3
**Rating:** 5
**Confidence:** 3

**Summary:**

The paper proposes a new type of noise for injecting in the latent dimension space to regularize the network. It shows that the popular Gaussian and Salt-Pepper noises are harmful and degrade performance. The paper shows the requirement for noise to have a positive effect. Based on this, a linear transformation noise is proposed, which is created by a circular shift Q matrix.

**Strengths:**

1. The proposed method is effective and well explained.
2. It works with popular architectures. CNN and Transformers show in the paper.
3. The method is helpful in domain adaptation as well.

**Weaknesses:**

1. Paper compares the proposed approach only with 2 basic latent dimension noise. Approaches similar to Manifold mixup should be compared: Noisy Feature Mixup
2. Why does the performance drop when combined with input augmentations (Table 17)? Input augmentations are staple of training neural networks. Incompatibility with it can be a big problem.
3. How does it compares against advanced input augmentation (like mixup, cutmix, randaug etc). Is better or worse? Is it compatible with them?
4. The paper's claim can be strengthened by showing results with another linear transformation noise.

**Questions:**

1. When compared to other noise for injection, the proposed approach shows beneficial. How does it fit with other regularization methods should be explored in more detail.
2. The paper comapres results when no input augmentation is present. How does the baseline methods compare with input augmentations?
3. Were the baseline noise arguments optimized? like the mean and variance of gaussian noise.
4. Domain adaptation results are compared with ViT and CNN based approaches but results are shown only with ViT based approach. Since, it is compared with CNN based approaches, it would be interesting to see how the approach performs with it.
5. How does it impact the training loss/accuracy?

---

### Official Review · Reviewer_MpsV · 2024-11-04

**Soundness:** 3
**Presentation:** 3
**Contribution:** 3
**Rating:** 6
**Confidence:** 3

**Summary:**

The paper explores how adding noise to deep neural networks can improve image classification and domain adaptation. It differentiates between positive noise, which helps, and harmful noise, which hurts model performance. The authors introduce NoisyNN, a method that uses positive noise to enhance model outcomes. Theoretical and empirical evidence supports the effectiveness of NoisyNN, showing it can achieve better results than traditional approaches. This study offers a new view on noise in machine learning.

**Strengths:**

- introduces a novel perspective on noise in deep learning by categorizing it into positive noise (PN) and harmful noise (HN) based on its impact on task complexity
- offers a fresh approach to understanding noise in the context of deep learning
- presents impressive experimental results

**Weaknesses:**

The results presented in the paper are excellent, and the authors have already provided the code. However, it is inconvenient to reproduce the experimental results without the pretrained models. Could you please provide models or APIs that would make it easier to replicate the results?

**Questions:**

Could you please provide models or APIs that would make it easier to replicate the results?

---

### Comment · Reviewer_xiNE · 2024-12-03
**Gratitude to AC for Script Support**

> [AC feedback] Even though any of us checked the implementation, it will not be possible to rigorously check the training dataset. Here is the checksum script for checking the validation split leakage (bash script). I assume that your training image directory looks like this: ..... (many scripts)

I sincerely thank AC for providing the Checksum script for verification. Across my experience as a reviewer in hundreds of papers, this marks the first time an AC has offered support at the script level. I deeply appreciate your expertise and commend this thoughtful gesture. Such actions contribute significantly to fostering a healthier and more robust research community.

> [author feedback] Thank AC for the suggestions. We ran the checksum script, there is a 797 overlap between training and validation images.

I appreciate the authors' honesty. That said, it is evident that this paper has substantial room for improvement overall. With more rigorous experiments, there could still be an opportunity to clarify its contributions and pursue publication.

Best,

Reviewer

---

### Meta-Review · Area_Chair_G4qF · 2024-12-18

**Metareview:**

This paper proposes NoisyNN, a regularization method that injects noise into the latent space. According to the paper, noise is typically harmful to neural networks, but noise can be positive under some conditions. Based on the derivation of the task entropy, this paper proposes to use positive linear transformation noises for neural network regularization. Experiments are conducted on image classification and domain adaptation.

This paper has mixed opinions, mostly negative ones. Also, the AC acknowledges that three out of four reviewers did not engage in the discussion, hence, the decision is not solely made by the average score.

The biggest problem of this paper is the reliability of the experimental results. As also pointed out by Reviewer xiNE, "Even with extensive web-scale pre-training, the best state-of-the-art performance remains below 92.5% (https://paperswithcode.com/sota/image-classification-on-imagenet)." I also agree that this paper reports suspicious experimental results (not only for 94.8% but also for numbers in Table 1-3). In my opinion, ResNet-50 with 81.3% top-1 accuracy is also suspicious. Even highly optimized ResNet-50 achieves 80.4% [A] top-1 accuracy -- Note that this model did not even follow the common training recipe, but was based on a highly optimized new recipe.

- [A] ResNet strikes back: An improved training procedure in timm

As mentioned in my previous comment, from my perspective, this paper should be validated very carefully and rigorously, including a thorough review of the implementation, trained weights, and related aspects. Given the unusually high numbers reported here, it is reasonable to expect the authors to demonstrate that their code and methodology are free of issues. However, as of now, I do not think that the experimental results are reliable. One possibility of the unusually high accuracies could be that noise injection is even performed during the inference, and the noise injection is performed batch-wise. There could be two potential bugs. (1) we do not shuffle mini-batch for validation, namely, the same mini-batch likely has the same classes (2) if the operation is affected by different samples in the mini-batch, then it uses other "test" samples for making a prediction. One possible simple validation will be to run the validation code with batch size 1. I tried to test this one while writing the meta-review, but unfortunately, the weight provided by the authors was not accessible during the meta-review phase.

There are also other concerns, such as the lack of comparisons with the other augmentations (vxMD, xiNE) the lack of implementation details (xiNE), the lack of instructions for the code (MpsV, xiNE), writing quality (xiNE). In my perspective, these concerns were not fully addressed because the reproducibility and reliability issues still remain.

I strongly suggest releasing the code publically based on `timm` library, which is the rule-of-thumb to ImageNet classification. If the result is truly reproducible, then it will be a great contribution to the community. However, if the implementation details are still unreproducible and hidden from open-source users, then I think the contribution of this submission will be limited.

As a minor comment, all the discussions between the authors and the reviewers are hidden from general audiences. The AC requested to change the visibility of the authors' comments to the authors, but the authors did not change the visibility. Although there is no specific policy for visibility (as far as I know), I think that open discussion is an important value to our community. I strongly recommend the authors change the visibility of the responses after decisions are made.

**Additional Comments On Reviewer Discussion:**

There were discussions between the authors and Reviewer xiNE. The reviewer raised a concern regarding reproducibility and reliability (e.g., a potential risk of test set leakage into the training set at a later training stage). Although the authors provided some details of the training script, as the reviewer mentioned, we can believe the results only if the same accuracy is achieved after the training is done with the same setting of the authors. However, as far as the AC understood, the results are still unreproducible and unreliable.

One possible reason could be the overlap between training and validation images. The AC provided a checksum script to check the overlap and the authors acknowledge an overlap between the training and test sets (797 images => note that it coincides with the known numbers by [A]), but the AC thinks the possible leakage can be happened in 21k images which is not tested during the rebuttal phase. Also, Reviewer xiNE mentioned that the authors do not update the codebase into the standard format, making it very challenging to reproduce it.

- [A] Section 4.5 of "When does dough become a bagel? Analyzing the remaining mistakes on ImageNet"

---

### Decision · Program_Chairs · 2025-01-22

Reject